# Different approaches to model the near shore circulation in the south shore of O'ahu, Hawaii

Joao Marcos Azevedo Correia de Souza[1,2] and Brian Powell[2]

[1]Centro de Investigacion Cientifica y de Educacion Superior de Ensenada, Baja California (CICESE). Carretera Ensenada-Tijuana No. 3918, Zona Playitas, C.P. 22860, Ensenada, B.C. Mexico.
[2]Department of Oceanography, University of Hawaii. 1000 Pope Rd., MSB, Honolulu, 96822 HI, USA.

*Correspondence to:* Joao M. A. C. Souza (jazevedo@cicese.mx)

**Abstract.** The dynamical interaction between currents, bathymetry, waves, and estuarian outflow have significant impacts on the surf-zone. We investigate the impacts of two strategies to include the effect of surface gravity waves on an ocean circulation model of the south shore of O'ahu, Hawaii. This area provides an ideal laboratory for the development of near shore circulation modeling systems for reef protected coastlines. We use two numerical models for circulation and waves: Regional Ocean Modeling System (ROMS) and Simulating Waves near shore (SWAN), respectively. The circulation model is nested within larger-scale models that capture the tidal, regional, and wind-forced circulation of the Hawaiian archipelago. Two strategies are explored for circulation modeling: forcing by the output of the wave model and online, two-way coupling of the circulation and wave models. In addition, the circulation model alone provides the reference for the circulation without the effect of the waves. These strategies are applied to two experiments: (1) typical trade-wind conditions that are frequent during summer months, and (2) the arrival of a large winter swell that wraps around the island. The results show the importance of considering the effect of the waves on the circulation and, particularly, the circulation-wave coupled processes. Both approaches show a similar near shore circulation pattern, with the presence of an offshore current in the middle beaches of Waikiki. Although the pattern of the offshore circulation remains the same, the coupled waves and circulation produce larger significant wave heights ($\approx 10\%$) and the formation of strong along- and cross-shore currents ($\approx 1$ m.s$^{-1}$).

## 1 Introduction

Our objective is to describe how ocean waves and currents interact in the south shore of the island of O'ahu, Hawaii, with a goal towards the development of an operational ocean forecast system. Much of the focus on ocean predictability has been at the larger scales within the ocean basins or on the continental slopes; however, human/ocean interaction is primarily within the near-shore surf-zone. Dynamical interplay between currents and bathymetry, currents and waves, ocean waters and estuaries, breaking waves, etc. may all significantly influence the predictability in the near-shore regions. In this paper, we investigate the impacts of surface gravity waves on the near-shore circulation in a high resolution regional ocean model for the coast of Honolulu, Hawaii (Figure 1). This work was developed under the umbrella of the Pacific Islands Ocean Observing System (PacIOOS) project (http://oos.soest.hawaii.edu/pacioos/), aiming to improve an operational coastal ocean forecast system for the island of O'ahu. The south shore of O'ahu is mostly contained within Mamala Bay including Waikiki beach. The challenge

is to provide useful forecasts of the near-shore circulation in this region that includes the primary dynamical processes while remaining feasible in computational cost. In the present work, alternatives are sought and tested for the development of such system.

In Mamala Bay, all scales of ocean dynamics are present from strong planetary mean flows, aperiodic mesoscale eddies impinging on the coastal region, internal tides, barotropic tides, and strong and variable trade-winds. The south shore of O'ahu within Mamala Bay is an ideal site for a case-study to understand the primary drivers of the near-shore circulation, quantify the particular contribution of the surface gravity waves to the circulation under different conditions, and determine what are the best options to represent such processes in operational forecast systems for exposed coral reef areas.

Among these processes influencing the near-shore circulation, Benetazzo et al. (2013) emphasizes the interaction between oceanic waves and currents as one of the main driving mechanisms for coastal regions. The authors show that wave-current interactions lead to important modifications of both the wave parameters—mainly wave significant height (Hs) and wave period—and the ocean currents in the Gulf of Venice. However, their bathymetry presents a gentle bottom slope. In the case of oceanic islands such as O'ahu, the steep slope and intricate bathymetry associated with coral reefs present an additional challenge for forecast systems.

The intermittent discharge of fresh waters from the Ala Wai canal (Figure 1c) can impact the near-shore density structure and ocean currents. On average, the influence of the canal on the coastal dynamics is small; however, sporadic, heavy rainfall events can force an outflow of freshwater into the coastal zone that alters the stratification. Although numerous modeling studies on large discharge rivers exist (Gracia Berdeal et al., 2002; Pan et al., 2014), the same is not true for rivers with small volume and aperiodic discharge.

In addition to the complicated bathymetry and the fresh water input from the Ala Wai canal, there are a number of processes that impact the near-shore currents near Waikiki. There are highly variable winds as the island mountains serve to create a wake region of lower but variable winds (Souza et al., 2015). Volcanic islands are not protected by wide shelves, which subject the near-shore to potential open ocean variability such as mesoscale eddies (Chavanne et al., 2010). Although lacking wide shelves, the islands are often protected by coral reefs that significantly alter the surface wave conditions and its interaction with ocean currents. Each of these phenomena can be significant in near-shore environments around the world, but Waikiki is an ideal laboratory for examining the influence since each of the varying dynamical scales are present. The implementation of a coupled circulation/wave model provides both the framework for a useful study on the theme and a forecast tool for operational purposes.

Hoeke et al. (2013) implemented a wave-circulation coupled model for the Hanalei Bay in the north shore of the island of Kauai, Hawaii (Figure 1a). Similar to the south shore of O'ahu, this region is characterized by a complex bathymetry, freshwater discharge, and surface waves that can dominate the dynamics. The authors used the Delft3D modeling system that combines the D-Flow circulation component with a wave component based on the Simulating Waves near-shore model (SWAN). The impact of waves on the circulation is calculated in the depth-averaged D-Flow momentum equations by including the wave induced forces as a source term. The enhanced bed shear stresses caused by waves are computed based on the Soulsby et al. (1993) formulation, and the wave forces are interpolated to the velocity points and substituted as explicit radiation stresses in

the momentum equation. However, an explicit description of the complex vertical fluxes of wave momentum is required to properly resolved the 3D circulation. This is particularly important for the wave induced mixing and the surf zone circulation. Lane et al. (2007) and Uchiyama et al. (2010) showed that the radiation stress approach used in the Delft3D system does not properly decompose the wave effects, and it obscures their underlying impact on the long (infragravity) waves and currents.
From the point of view of the wave field, Edwards et al. (2009) show the Delft3D system tends to underestimate the wave height.

Lowe et al. (2009) applied the same model system as Hoeke et al. (2013) to study the circulation in the coastal reef-lagoon system of Kaneohe Bay in the northeast coast of the island of O'ahu, Hawaii (Figure 1b). Most previous studies assume the wave setup — the local increase in the sea level due to the wave breaking — in the reef lagoon to be negligible, as this is
common for ocean atolls and barrier reefs. However, the authors emphasize the fact that, if the water exchange is restricted to relatively narrow channels in the reef as in Kaneohe Bay and Waikiki, a water level difference between the lagoon and the open ocean will be present and will establish a pressure gradient impacting the local circulation. The authors did not consider the effect of freshwater input from river discharges that influence the lagoon circulation.

While focusing on the fate of harmful bacteria from the Ala Wai plume, Johnson et al. (2013) developed a Regional Ocean
Modeling System (ROMS) simulation for a similar region as this study. They examined sporadic events of large flux from the Ala Wai canal using a relatively coarse horizontal resolution ( 250m) model incapable of resolving the several channels in the reef banks. As shown by Plant et al. (2009) the resolved wave height is very sensitive to the cross-shore bathymetry resolution, while the resolved currents are more sensitive to the along-shore resolution. Comparing model results to observations, these authors demonstrate how the errors in the modeled currents increase if one uses smoothing scales larger than 100m for the
bathymetry. To examine the influence of waves, Johnson et al. (2013) used a prescribed wave field as additional forcing to the ROMS model that did not consider the influence of the currents on the waves. Moreover, the authors used a parameterization based on the Mellor (2003, 2005) approach to estimate the modification of the currents by the wave field. In addition to the radiation stress approach problems discussed, this particular method is known to generate inconsistent pressure fields as demonstrated by Ardhuin et al. (2008).
Therefore, a different approach to the wave-current interaction off the south shore of O'ahu is necessary. This paper aims to clarify: (i) the effective contribution from the surface waves under different conditions; (ii) the importance of coupled ocean currents and surface wave processes for the local dynamics; and (iii) the influence of different approaches for forecasting the near-shore current field.

This research uses a suite of numerical models to examine these main questions, as there is little observational data available.
Although the Kilo Nalu cabled reef observatory once provided real-time observations of several physical and biogeochemical parameters Samsone et al. (2008), the lack of continuous measurements of the near shore currents in Honolulu during the modeled period (and in general) makes it impossible to properly validate the results and quantify the performance of each modeling strategy. Unfortunately, there are no results from the atmospheric model and the ocean model system used to provide surface forcing and boundary conditions to the near shore domain during the period of the Kilo Nalu experiment. However,
contrasting the results from the different model strategies helps to reveal robust circulation patterns and clarify what differences

should be expected when adopting each method on a modeling system. A qualitative analysis is performed to understand the modifications on the near shore circulation. This can assist in the future development of near shore forecasting systems. It is important to note that this study was part of the design of the operational near shore forecast system for the south shore of O'ahu. This system uses the model setup presented here to provide daily forecasts of waves and currents for the Honolulu
shore.

## 2 Methods

### 2.1 The ROMS model

The ROMS is a 3D primitive equations ocean model using hydrostatic and Boussinesq approximations. A full description of the model can be found in Shchepetkin and McWilliams (2005); McWilliams (2009) and the ROMS website (www.myroms.org).
We make use of the Coupled-Ocean-Atmosphere-Wave-Sediment Transport Modeling System (COAWST) described by Warner et al. (2010), that provides on-line, two-way coupling between ROMS, SWAN, and Weather Regional Forecast (WRF) models through the Model Coupling Toolkit (MCT). We implemented the coupled ROMS / SWAN simulations for the south Shore of Honolulu, Hawaii using a vortex force formalism to account for the wave-current interaction described by Uchiyama et al. (2010) and Kumar et al. (2012), which gives better performance than the traditional Mellor (2005, 2008) radiation stress ap-
proach (Lane et al., 2007). All simulations use the Mellor and Yamada (1982) turbulence closure model to account for the vertical mixing.

We utilize a horizontally variable grid with ≈50m resolution in the region between Waikiki and the Honolulu Harbor, gradually decreasing to ≈100m at the boundaries (Figure 1c), and it covers a total length of 11.5km alongshore and 4.5km offshore. Therefore the grid is rectangular, with a variable horizontal resolution and rotated to fit O'ahu's south shore orien-
tation. Since the grid is oriented with the mean coastline, the model directly outputs along shore and cross shore velocities. Although these velocities are not perfectly oriented parallel and perpendicular to the coast at every individual point, it provides the correct components when considering the regional circulation pattern. There are 10 vertical layers. Since much of the domain is less than 10m deep, in most areas the vertical resolution is less than 1m. It is interesting to note that only 18% of the model water grid cells are over 50m deep (4% over 100m with all deep cells concentrated at the southern boundary.
The maximum depth of the domain is 300m, in the southeast corner of the model grid. The minimum depth of the grid water points is of 0.5m to account for the shallow reef areas. This grid is sequentially nested within three ROMS circulation models of approximately 250m, 1km, and 4km resolutions that span from the south shore to the entirety of the Hawaiian islands (not presented).

The open boundaries of the near shore domain are forced with barotropic tides and circulation, temperature and salinity
from the coarser 250m parent-grid. The horizontal resolution minimizes the errors in the resolved circulation, as shown in comparisons with observations by Plant et al. (2009). All of the grids are part of an operational forecast system, the Pacific Islands Ocean Observing System (PacIOOS - http://pacioos.org). The outer grids are run using an incremental strong constraint four-dimensional variational data assimilation scheme, as described by Matthews et al. (2012). A description and evaluation of

the assimilation system is provided by Souza et al. (2015). The present simulations were nested inside the ≈250m horizontal resolution grid comprehending the south Shore of O'ahu, the same models used by Johnson et al. (2013), with boundary conditions provided every 3 hours. A buffer zone gradually merges the two grids in terms of resolution and bathymetry. In the PacIOOS project all simulations are nested offline due to operational reasons, such that the small scale processes in the inner grids have no impact on the outer grids.

The models use surface forcing fields from a locally produced WRF model run performed under the PacIOOS project. Eleven tidal constituents obtained from the Oregon State University TOPEX/Poseidon Global Inverse Solution (TPXO) (Egbert and Erofeeva, 2002) were introduced as a separate spectral forcing in the outer grids (Janekovic and Powell, 2011). Through this method the tides are introduced in the barotropic velocities and elevation at the open boundaries and as tidal potential at every grid point (amplitude and phase of 11 tide constituents provided at each grid point). Since the TPXO does not properly represent the tides in near shore shallow regions with complicate bathymetry and geography, the results from the outer grids were used to generate similar spectral tidal forcing for the near shore domain. Internal tides generated in the outer grids are directly introduced to the inner grids through the baroclinic fields at the boundaries. Input fluxes from the Kalihi and Palolo-Manoa channels were obtained from the USGS (http://waterdata.usgs.gov). Similar to Johnson et al. (2013), the Palolo-Manoa channel flux was multiplied by 1.3 to account for the contributions from runoff waters and smaller drainage sources into the Ala-Wai canal.

As described by Kumar et al. (2012), the effect of the waves on the circulation is expressed in the inclusion of new terms in the right hand side of the models governing equations:

$$\frac{\partial \mathbf{u}}{\partial t} + (\mathbf{u}.\nabla_\perp)\mathbf{u} + w\frac{\partial \mathbf{u}}{\partial z} + f\hat{z} \times \mathbf{u} + \nabla_\perp \varphi - \mathbf{F} - \mathbf{D} + \frac{\partial}{\partial z}(\overline{u'w'} - v\frac{\partial \mathbf{u}}{\partial z})$$
$$= -\nabla_\perp \mathscr{K} + \mathbf{J} + \mathbf{F^w}$$
$$\frac{\partial \varphi}{\partial z} + \frac{g\rho}{\rho_0} = -\frac{\partial \mathscr{K}}{\partial z} + K$$
$$\nabla_\perp.\mathbf{u} + \frac{\partial w}{\partial z} = 0 \tag{1}$$

where $(\mathbf{u},w)$ are the Eulerian mean horizontal and vertical velocities, $\nabla_\perp$ is the horizontal differential operator, $\varphi$ is the dynamic pressure normalized by the density, $\mathscr{K}$ is the lower order Bernoulli head, $(\mathbf{J},K)$ is the vortex force, and $\mathbf{F}^w$ is the sum of the momentum flux due to the non-conservative wave forces. The continuity equation is included for completeness, and the tracer equation is not presented. It is important to note that we will refer to the quasi-Eulerian mean velocities as the horizontal currents. As defined by Kumar et al. (2012), this velocity is the Lagrangian mean velocity minus Stokes drift. This is the velocity output by the COAWST system.

As described by Uchiyama et al. (2010), the Stokes drift velocities are defined by:

$$\mathbf{u}^{st} = \frac{A^2\sigma}{2sinh^2[\mathscr{H}]}cosh[2\mathscr{L}]\mathbf{k}$$
$$\omega^{st} = -\nabla_\perp.\int \mathbf{u}^{st}dz', \tag{2}$$

where $(\mathbf{u}^{st}, \omega^{st})$ are the 3D non-divergent Stokes velocities, $A$ is the wave amplitude, $\sigma$ is the intrinsic frequency, and $\mathscr{H}$ and $\mathscr{L}$ are the normalized vertical lengths. The Stokes velocity is proportional to the squared wave amplitude, with a smaller influence of the wave height on $\mathscr{H}$.

## 2.2 The SWAN model

SWAN is a third-generation spectral wave model developed at the Delft University of Techonology. It solves the spectral action density balance to describe the evolution of wave energy over direction and frequency, time, and space. It is able to resolve the wave generation by winds, energy transfer by the wave-wave interactions, shoaling and refraction due to the bathymetry and currents, and wave dissipation by white capping, bottom friction, and breaking in the near shore area. It has been a proven tool for modeling complex wave fields in coastal regions with the varying bathymetry and in the presence of complex currents.

The model was developed by Booij et al. (1999) and provides an efficient solution for modeling near-shore waves. The action balance equation describes the evolution of the wave action spectrum, *N*, as:

$$\frac{\partial N}{\partial t} + \frac{\partial c_x N}{\partial x} + \frac{\partial c_y N}{\partial y} + \frac{\partial c_\sigma N}{\partial \sigma} + \frac{\partial c_\theta N}{\partial \theta} = \frac{S}{\theta}, \tag{3}$$

where $t$ is time, (x,y) are Cartesian coordinates, $(c_x, c_y)$ are the propagation velocities of wave energy in x and y, $\theta$ is the wave direction, $\sigma$ is the wave frequency, $c_\theta$ and $c_\sigma$ are the propagation velocities in spectral space $(\theta, \sigma)$, and $S$ represents the source

terms. The parameterizations in the source terms cater to the wave processes from deep to intermediate water depth, which include wind-wave interactions, quadratic wave interaction, dissipation due to white capping, bottom friction as well as the coastal wave processes including refraction due to a current field, triad wave interactions, and depth-induced wave breaking. Due to its ability to account for the wave-current interaction, SWAN coupled with other circulation models is suitable for near-shore hydrodynamic studies.

With the same computational grid as the near shore ROMS simulations shown in Figure 1c, the near shore SWAN wave model was forced by the same high resolution WRF wind used by the circulation model, available from PacIOOS. The Hawaiian wave forecast system in the same PacIOOS project outputs the 2D spectra boundary condition for the SWAN domain to calculate the wave transformation in the coast of Honolulu. The SWAN spectrum is discretized by 24 equal directional bins from $0°$ to $360°$ and 25 exponentially increasing frequency bins from 0.0418 to 1 Hz on each grid. The spectral density over

the domain was updated every 5min during the wave modeling, and wave parameters were output every hour.

As described by Booij et al. (1999), triad wave-wave interactions and depth-induced wave breaking are parameterized using the Eldeberky and Battjes (1995) and Battjes and Janssen (1978) models respectively. For more details, please see the Appendix in Booij et al. (1999).

## 2.3 The coupling process

The models were coupled via the Model Coupling Toolkit (MCT), that allows the exchange of information between the ocean and wave models. This exchange of information is independent from each model's grid and time step, and we a use coupling

time step of 120s. The coupling time step is a compromise between the computational cost and the time scale of the variability of the properties exchanged by the models. Although sensitivity tests showed that a 1 hour time step would be sufficient, a more conservative approach was adopted.

As described by Warner et al. (2010), at each coupling time step the wave model (SWAN) provides results on wave height, wave length, wave direction, surface and bottom periods, percent waves breaking, bottom orbital velocity, and wave energy dissipation to the ocean model (ROMS). At the same time, ROMS provides the near-surface currents integrated over a depth of one wave height, free surface elevation, and bathymetry (constant in time for our case). The exchange of mean wave parameters between models assumes the wave field is dominated by a well defined sea swell, which was found to be a reasonable approach for O'ahu's south shore.

## 2.4   Model experiments

Three groups of simulations were designed to study the impact of the surface gravity waves on the currents: (1) standalone ROMS model without considering the waves (**NOWAVE**); (2) ROMS model including hourly forcing from SWAN (**WAVEFORCE**); and (3) two-way coupled ROMS/SWAN simulations (**WAVECOUPLE**) using the MCT.

Each of these simulations were run for two, 5 days experimental periods with different wave conditions:

Experiment 1  09/08/2013 to 09/13/2013 — Moderate south waves ( 1m) and evening rains corresponding to typical boreal autumn conditions. The evening rains translate into fresh water pulses in the river fluxes, particularly from the Ala Wai canal.

Experiment 2  01/21/2014 to 01/26/2014 — Large north swell that wraps around the island with the presence of southeast swell, generating waves above 2m in the surf zone in Honolulu. A relatively large rain event occurred in the evening of the second day of simulation; however, the river fluxes are over 30 times smaller than previously observed extreme events, such as the case study presented by Johnson et al. (2013). This means the influence of the river discharge on the water column stratification is restricted to the Ala Wai mouth.

The wave conditions were chosen based on 2 years of data from the Kilo Nalu observatory(Samsone et al., 2008) presented in Figure 2. This data set shows a predominance of small (under 1m) waves from the south in the south shore of Honolulu. A few larger events are present in the data set, with very few variability in the direction. This is to be expected since the observatory was located very close to the shore line and the waves stir to a direction perpendicular to the bathymetry as approaching the shore. Unfortunately the Kilo Nalu observatory was installed northwest of the Waikiki beach, and was not operational for the period of the present study.

Despite the difference in the time step at which the wave parameters are provided to the circulation model in the WAVE-FORCE (1 hour) and WAVECOUPLE (2 min.) approaches, it does not impact the model solutions. This relates to the slow pace of change of the wave characteristics for both experiments, that are evident in the Stokes drift velocity kinetic energy time series of Fig. 8e.

## 3  Results and Discussion

### 3.1  Model evaluation

NOAA National Database Buoy Center (NDBC) buoys shown in Figure 1 provide continuous measurement of the wave parameters around Hawaii. Despite their locations far from the south shore of O'ahu, the wave records at these buoys provide an

evaluation for the wave hindcast system. Since no observations are available inside the near shore domain, we proceed with a comparison of the NDBC buoys with the outer SWAN grid (SWAN only) that provide the boundary conditions to the inner domain. Figure 3 shows good agreement between the measured and modeled significant wave height, peak period, and peak wave direction at offshore buoy 51003 and near shore buoy 51201, 51202, 51203, and 52104 for the time period of experiment 1. The wave conditions at buoy 51003 indicate dominant trade wind generated waves from the east with wave height below 2.5

m. An intermittent north swell with peak period of 15 s is evident at buoy 51201, 51202 for Sep. 11. Buoys 51203 and 51204 sheltered from east wind waves and north swells show mild south swells with significant wave height less than 1 m. The wave hindcast provides a useful tool to reproduce the multi-model waves conditions in Hawaii.

In contrast, both the wave hindcast and measurements in the near shore buoys 51201, 51202, 51203, 51204 show large northwest swells with peak period above 15 s for experiment 2 in Figure 4. The peak wave height decreases from 7 m at buoy

51201 in the north shore to 5 m at buoy 51204 in the south shore of O'ahu. Although no wave data is available in Waikiki for the time periods of experiment 1 and 2, the agreement between the outer wave model and the NDBC buoy records indicate that representative boundary conditions are provided to the O'ahu south shore domain. Therefore, based on the comparisons presented in Figures 3 and 4, we confirm the skill of the spectral wave model to represent the typical multi-model waves conditions in experiment 1 as well as the arrival of the large swells in experiment 2.

In general, the root-mean-square deviations (rmsd) show a good agreement between the model results and the buoy data (Table 1). Special attention should be taken on the buoy 51204 - the closest buoy to the model domain and the only one in the south side of O'ahu. Further analysis of the model system performance based on 34 years of model hindcasts is provided by Li et al. (2016).

Similar to the waves, there is no data available on the near shore currents in the south shore of O'ahu during the experiments

period. Nevertheless, the model system that provides the boundary conditions of the coastal domain has been validated against satellite sea level anomaly, sea surface temperature, and high-frequency radar surface currents (Souza et al., 2015). The authors calculate root-mean-square errors (rmse) for the outer grid of 0.05±0.06m for SSH, 0.25±0.25°C for temperature and 6.5±6cm/s for surface velocities, when compared to along track SSH, temperature from Modis satellite and Argo profiles, and radial velocities from and High Frequency Radar (HFR) observations.

The HFR data gives widespread surface currents data for the south shore of O'ahu - the region of interest of the present study. Although it does not provide observations of the near shore circulation, it does provide an important constrain to the outer grid surface velocities used as boundary conditions to the near shore domain. In their figures 2 and 3, Souza et al. (2015) show a good agreement of the modeled radial velocities with the observations for the south shore of O'ahu with rmse generally

under 10cm/s for their experiment A (equivalent to the present model setup). The authors calculate a phase difference of 5.2 min for the $M_2$ tide component - the main tide component for this region.

## 3.2 Effects on the waves

The WAVECOUPLE simulation has longer periods (T) throughout the domain and larger Hs than the WAVEFORCE in the reef region, providing an indication to the effect of the circulation on the waves as described by Lowe et al. (2009) (Fig 5). According to Warner et al. (2010) the ocean currents affect the modeled waves by modifying the wind stress and the group velocities, $\mathbf{c} = (c_x, c_y)$. Since the model uses bulk formulas to calculate the wind stress, it will reflect the modification of the 10m winds $\mathbf{u}_{wind}$ by the moving ocean surface $\mathbf{u}_{wind} - \mathbf{u}$, where $\mathbf{u}$ is the surface current velocity.

The ocean currents modify the group velocities $\mathbf{c}$ by adding the current: $\mathbf{c} + \mathbf{u}$. This alters the wave number and allows for current-induced reflection as shown by Fan et al. (2009). The authors showed that the interaction with the ocean currents causes a Doppler shift for the gravity waves. When the currents are against the waves, the waves are compressed, and when the currents have the same direction as the waves, the waves are elongated. The degree to which this Doppler shift modifies the surface waves depends on the current speed and direction relative to the wave propagation speed and direction; therefore, short waves with slow propagation are most affected by the ocean currents.

Following Oh and Kim (1992) the currents affect the wave's apparent period, $\omega$, through:

$$\omega = kU(\boldsymbol{x}, t) + \sigma(k, h) \tag{4}$$

where $k$ is the wave number, $U$ is the current intensity along the wave's propagation direction, $\sigma$ is the intrinsic frequency, and $h$ is the local water depth.

The spatial variability of the differences in Hs, direction and period (T) between the coupled and forced simulations (WAVE-COUPLE - WAVEFORCE) are shown in Figure 5. Waikiki is characterized by the presence of a coral reef system and complicated bathymetry with strong currents. Due to its complexity and impact for the local community, the following analysis will focus on the region of Waikiki.

WAVECOUPLE exhibits $\approx 10\%$ higher Hs averaged in Waikiki area than WAVEFORCE, with differences concentrated in the reef zone. Comparing these difference maps to the bathymetry in Figure 1c and the time averaged currents for the NOWAVE cases in Figures 6(a,b) and /reff6(a,b), one can note that the differences in Hs are concentrated over shallow reef areas, while differences in wave direction have a more widespread distribution. As expected, the magnification of Hs due to the coupling occurs in the western and eastern extremes of Waikiki, causing increases in the mean water level due to wave setup at the coast.

In the return flow area in the middle of bay formed by the Waikiki beach (see Figure 1c), there is almost no change in Hs. This area corresponds to a large channel in the reef (see Fig. 1c), where there is very low depth induced wave breaking, as will be seen in the next section.

To analyze the effect of the circulation on the waves the differences in wave direction and Hs between the WAVECOUPLE and WAVEFORCE simulations were correlated to the current intensity and direction for both experiments (Table 1). For that, time series of the differences and current direction and intensity in each grid point in the domain covered in Figure 5 were

extracted and the obtained correlations spatially averaged. While Experiment 1 exhibits higher correlations with the current direction, Experiment 2 was more correlated to the current intensity. Since the mean difference between the waves and currents direction is similar for both experiments ($\approx$60 degrees), this distinction cannot be explained by the Doppler shift discussed above. It appears the correlation is a function of the relation between the wave and currents intensity. Comparing Experiments 1 and 2 for both modeling strategies shows that, while Experiment 1 presents stronger ocean currents (by 90%), Experiment 2 exhibits larger wave heights (by 15%). The smaller waves in Experiment 1 are subject to the influence of the stronger currents explaining the higher impact of the coupling. The fact the NDBC data shows a prevalence of small waves (Hs<1m 94% of the time) emphasizes the importance of the interaction with the local currents.

## 3.3 Effects on the currents

Although the effect of the Ala Wai canal discharge in the near-shore circulation can be significant during large rain events (Johnson et al., 2013), we focus on the effects of the waves in periods when the Ala Wai low volume flux does not impact the local circulation.

The resulting mean circulations obtained from the three modeling strategies (Figures 6 and 7) clearly show the effects of the waves and the wave/current interaction on the resolved circulation. The near-shore current pattern drastically changes when introducing the effect of the waves, while the offshore currents keep their general spatial structure (albeit with different intensity). The formation of local circulation cells in the near-shore is related to the presence of return flows that are forced by the pressure-gradient due to the wave setup.

The along-shore component of the velocity (Figure 6 ) shows the formation of coastal drift currents in both WAVEFORCE and WAVECOUPLE simulations. For the remaining of the text we refer to along-shore as along the mean shoreline orientation. A similar reasoning is used for the cross-shore velocity component. The pattern shows the convergence of this flow in the central area of Waikiki, with the WAVECOUPLE exhibiting larger intensities. Similarly, Figure 7 presents the cross-shore component of the surface velocity. The formation of onshore/offshore flow cells is evident in the figure, with the WAVECOUPLE exhibiting intensified flow. A strong negative (offshore) current is present in the area of convergence of the coastal drift. This corresponds to a strong near-shore circulation cell with the presence of intensified cross-shore current. In Experiment 1, several smaller cross-shore current cells are present in the western portion of Waikiki with onshore currents over the reef heads and offshore currents in the small channels in the reef. Experiment 2 shows a pattern dominated by an unique offshore flow at the convergence area with positive onshore flows both to the east and west. The eastern part of Waikiki is dominated by a strong northwest flow in both experiments that is independent of the modeling strategy. This is related to the fact that—independent of the direction of the incident swell—the near shore waves have similar direction when approaching the coastline, and they break over the shallow reef close to the beach in the east portion of Waikiki. A direct comparison between both modeling approaches in Figures 6 and 7 demonstrates that both reproduce the same near-shore circulation pattern but with different intensities.

The modification of the circulation by the waves is expressed by the right-hand side terms in Eq. (1). It includes the influences of the vortex force, Bernoulli head, and non-conservative wave forces. These wave effects enter the ROMS primitive equations as momentum and tracer fluxes. The vortex force (VF) terms represent the interaction between the Stokes drift and the vorticity

of the mean flow. Since this term is not explicitly written in the model output, it is difficult to quantify its contribution to the momentum balance. Nevertheless, it is directly related to the Stokes drift velocities obtained from the resolved wave field, as expressed by Benetazzo et al. (2013):

$$\mathbf{VF}_{hor} = -\hat{z} \times \mathbf{u}^{st}(\hat{z}.\nabla_\perp \times \mathbf{u} + f) - w^{st}\frac{\partial \mathbf{u}}{\partial z} \tag{5}$$

where $\mathbf{VF}_{hor}$ is the horizontal component of the vortex force, and $\hat{z}$ is the vertical unit vector.

The maps in Figure 8 (a to d) show the spatial distribution of the surface Stokes velocities in the Waikiki area. The large signal in the kinetic energy in Experiment 2 shows the arrival of the large swell on January 22th. Although the WAVEFORCE simulations exhibit slightly higher Stokes current intensity throughout the domain, the shallow near-shore region over the reef (under 10m depth) have stronger Stokes currents in the WAVECOUPLE simulations associated with the magnification of the

waves in the surf zone due to the interaction with the currents. This is evident in the time series of the kinetic energy associated with the Stokes velocities in the Figure 8e and f. The Stokes drift velocities are nearly opposite to the mean circulation resolved by the NOWAVE simulations (Figures 6 and 7 a and b). This explains the smaller average velocities and the smaller total kinetic energy resolved by the WAVECOUPLE simulations in the offshore region. While for Experiment 1 both modeling strategies show similar Stokes velocity kinetic energy time variability, Experiment 2 exhibits a clear peak associated to the arrival of the

swell.

This difference in the Stokes drift velocities, however, is not enough to explain the observed differences in the total currents. Taking only the wave effects into consideration, the differences in the total velocity intensities are mainly a consequence of the wave setup/setdown. The presence of cross-shore current cells is the main feature in the velocity maps of Figure 7. These circulation cells are a consequence of mean sea level increases (wave setup) shoreward of the wave breaking zone, generating a

pressure gradient that balances the radiation stress. As pointed out by Dalrymple et al. (2011), cross-shore currents are usually generated simply by alongshore variations in breaking wave heights. In bays such as Waikiki, cross-shore currents can form in the center of the beach and extend significantly offshore. The models results indicate cross-shore currents exceeding $1\,\mathrm{m\,s^{-1}}$ in the near shore region in Waikiki for both WAVECOUPLE and WAVEFORCE simulations, as typically observed for atolls and barrier reefs according to Gourlay and Colleter (2005). The differences between the two experiments show that the intensity

and duration of the high wave event dominate the circulation response in the Experiment 2. While in Experiment 1 the wave regime is quasi-constant and the near shore circulation is in balance, the arrival of a large swell in Experiment 2 perturbs the balance and causes a stronger circulation response in the WAVECOUPLE approach. The modification of the wave direction due to the interaction with the currents acts to modify the setup and generate a feedback on the currents.

Therefore, it is necessary to quantify the modification of the sea-level by the waves and the balance with the dissipation of

wave energy. To achieve this, the Sea Surface Height (SSH) differences between the simulations that include the effects of the waves (WAVEFORCE and WAVECOUPLE) and the NOWAVE were calculated and are presented in Figure 9. Although these differences are small (order 1cm), the consequences on the near shore circulation are important (as shown in the near shore currents pattern in Figures 6 and 7). Figure 10 reveals the differences in the cross-shore pressure gradient. From here, we will present the pressure gradient as a per density unit.

The elevation of the sea surface near the coast due to the waves is observed for each experiment, followed by a lowering of the sea level towards the open ocean. The WAVECOUPLE cases show overall larger magnitude of elevation, both positive and negative, than the WAVEFORCE. These differences in the sea level impose a cross-shore pressure gradient that affect the local currents.

Taking the cross-shore section (1) shown in Figure 9d as an example, Figure 10 shows the pressure gradient obtained by each modeling strategy for both experiments with the associated cross-shore surface velocities. The larger shaded area in Figure 10b and d in relation to 10a and c reflects the arrival of the larger swell waves in Experiment 2, and the consequent increase of the cross-shore pressure gradient and velocity. There is a difference in the spatial distribution of the pressure gradient between the WAVEFORCE and the WAVECOUPLE simulations. Both maxima are aligned to the reef break, but the WAVECOUPLE simulation shows an overall smoother transition towards the coast and offshore, with smaller gradients in the pressure that reflects the larger and broader wave setup as observed in Figure 9. The cross-shore velocities follow a similar pattern, mirroring the pressure gradient sections. There is a shift in the maxima between the WAVECOUPLE and WAVEFORCE that reflects the smoother transition of the wave setup observed in the pressure gradient sections. The feedback circle is closed when the Doppler shift by the near-shore currents modify the wave field that generates the pressure gradient against the shoreline. This generates a modified cross-shore pressure gradient affecting the currents and closing the feedback. In the WAVEFORCE experiments this feedback is broken because only half of it is represented by the model dynamics.

As explained by Kumar et al. (2012), there is a balance between the wave setup derived pressure gradient and part of the wave energy dissipation that contributes to the momentum flux in the surf zone. This dissipation is part of the non-conservative wave forcing in Eq. (1), that includes depth-induced wave breaking (and white capping) near the surface and frictional wave dissipation near the bottom. The remaining energy from the wave dissipation is involved in the creation of wave rollers, that are related to turbulent mixing in the surf zone and dissipation. Figure 11 shows the dissipation by wave breaking for the WAVECOUPLE simulations, that is $\approx 10^2$ larger than the dissipation by white capping and $\approx 10^6$ greater than the dissipation through bottom friction. The distribution of dissipation by wave breaking is not uniform along the Waikiki beach. There is a region of low dissipation in the middle of the beach, corresponding to the intense return flow observed in Fig. 7. The energy dissipation for Experiment 2 is higher than Experiment 1 because it contains larger wave heights.

To analyze how the different phenomena interact to generate the observed near-shore circulation pattern, the pressure gradient (cross- and along-shore), the non-conservative wave forces (sum of depth induced breaking, white capping, and bottom friction), and the integrated cross-shore transport by the quasi-Eulerian currents and the Stokes drift are plotted for the two sections shown in Fig. 9d. To isolate the contribution of the wave setup to the surface currents, the velocities obtained by the NOWAVE simulations were subtracted from the total cross-shore velocities prior to the transport calculation. The WAVECOU-PLE Experiment 2 was taken to demonstrate the differences in the balance between the two sections. The results are presented in Fig. 12.

At the section (1) (Fig. 12a) there is a correspondence between the maxima of the non-conservative wave forces on the reef edge, the negative (offshore) pressure gradient, and the onshore water transport due to the Stokes drift currents in the surf zone. The cross-shore transport is insensitive to this balance in the surf zone. The wave energy dissipates rapidly to zero more than

≈400m offshore as the cross-shore pressure gradient becomes positive (onshore), indicating the wave shoaling region where setdown (reduction of the mean sea level, Fig. 8) takes place. This is a good example of how the energy from the breaking waves drive the currents near the coast. As explained by Dalrymple et al. (2011), at the offshore edge of the surf zone the waves steepen and break, propagate across the surf zone, and run up the beach. Balancing forces are required for the energy loss by wave breaking and consequent change in the cross-shore and/or alongshore momentum flux. These primarily arise from wave induced changes in the mean water level at the shoreline (wave setup) that provides a hydrostatic force due to wave-induced currents.

There is a large channel in the reef near the region of section (2). As a consequence of the larger depths close to the coastline, the energy loss by wave breaking is approximately 6 times smaller than in section (1) and concentrated near the coast (shorebreak). The balance explained above for section (1) does not take place, as is evident by the lack of associated maxima in the cross-shore pressure gradient and Stokes drift velocities transport. This section, however, is in the convergence zone of alongshore wave induced currents observed in Fig. 6. A strong (3 times larger than in section 1) cross-shore transport is generated as the return branch of the near-shore circulation cell. The local along-shore pressure gradient show large values, ranging from negative close to the beach to positive offshore. The water transport seems to respond to the larger scale (order of the beach length) pressure gradient that is evident from the wave setup maps in Fig. 9.

Revisiting the cross-shore velocity maps in Fig. 7 and comparing with the non-conservative wave forces of Fig. 10, there is a clear difference between the small wave condition (≈ 1m) of Experiment 1 and the arrival of a large swell in Experiment 2. In Experiment 1, there is the formation of several small, cross-shore cells with a stronger one in the region of along-shore current convergence at the middle of the beach where section (2) is located. For Experiment 2, wave breaking dominates the western portion of Waikiki with a stronger wave setup, larger convergence at the middle of the beach, and consequent stronger cross-shore currents in the region of section (2).

With the mechanism of cross-shore circulation cells in mind, it becomes clear how the small observed differences in the pressure gradient associated to the wave setup/setdown resolved by the coupled simulations have important impacts on the near-shore circulation. This demonstrates how significant coupled processes are for the resolved currents and for the skill of a near-shore forecast systems.

To quantify these differences, Table 2 presents a comparison of the total velocity, Stokes drift, and setup associated velocity between the WAVECOUPLE and WAVEFORCE simulations. The setup associated velocities were calculated by subtracting the NOWAVE velocities from the model quasi-Eulerian velocities. The NOWAVE is taken to represent all of the other contributions not associated to the waves action. The larger intensity of the Stokes drift in the WAVECOUPLE simulations is related to the larger Hs (≈10%) in relation to the WAVEFORCE simulations for both Experiments.

Therefore, the interaction between the surface waves and near-shore circulation have important impacts on the resolved currents, especially in the near-shore region between the reef crest and the coast. This is an important phenomenon that should be taken into consideration when developing forecast systems that aim to provide a useful description of the near-shore currents.

## 4 Summary and Conclusions

Due to the interaction with the currents that modify the wave number, coupling the circulation and wave models give rise to larger Hs and slightly different wave directions when compared to the SWAN alone. The smaller waves present in Experiment 1—the type that prevail most the time in the south shore of O'ahu—are overall more sensitive to the local circulation. While the differences in the Hs are concentrated in the reef region, the modification of the directions due to the interaction with the currents is widespread through the domain.

The differences in the resolved wave fields presented feedbacks on the circulation in the coupled simulations (both WAVE-FORCE and WAVECOUPLE), since the resolved Hs reflects in distinct Stokes drift velocities and wave setup against the coast. Such differences are magnified in the reef crest area and in the region between the reef and the coast. The Stokes drift velocities flow opposite to the surface currents resolved by the NOWAVE case, resulting in weaker currents when considering the effect of the waves (both WAVEFORCE and WAVECOUPLE approaches). The wave setup determined the free-surface elevation on the reef and dominates the near-shore circulation, similar to the case of Kaneohe Bay studied by Lowe et al. (2009). This effect was observed to be stronger in the occurrence of a large swell event in the Experiment 2. The wave setup was the dominant process determining how the waves affect the near-shore circulation in the Honolulu coast. This conclusion should be transferable to other exposed coral reef coastal areas, where the near-shore circulation is deeply affected by the surface gravity waves particularly in the occurrence of large swell events.

In this paper, we set out to understand: (i) the effective contribution from the surface waves under different conditions; (ii) the importance of coupled ocean currents and surface wave processes for the local dynamics; and (iii) the influence of different approaches for forecasting the near-shore current field.

We found that the surface wave field has significant importance to the circulation, even in periods of small (under 1m) waves. (i) The consequences on the Stokes drift velocities, wave setup, and non-conservative wave forces influence the circulation through the domain, with a particularly important contribution in the surf zone on the reef break. (ii) Although the general near-shore circulation pattern is resolved in both WAVEFORCE and WAVECOUPLE simulations, the inclusion of coupled processes led to differences in the magnitude of the currents and wave parameters. There were important differences in the cross-shore circulation cells resolved by the two approaches, generally with stronger cross-shore currents for the WAVECOUPLE simulations. The inclusion of coupled processes was shown to be important in the representation of both wave and circulation parameters, leading to overall larger Hs (≈10%), longer wave periods (≈30%), and associated stronger Stokes drift currents. (iii) Despite the improvements the coupling may bring, one must also consider the computational cost of the numerical simulations — especially for high resolution coastal grids. Coupled simulations are extremely expensive, and the WAVEFORCE approach can provide an interesting alternative.

The results show the importance of considering coupled processes when aiming to resolving both the near-shore circulation and the waves characteristics in the reef zone. However, the computational cost involved in coupled simulations presents an important obstacle in the use of this approach for operational forecast systems. Once in possession of the wave model results, the WAVEFORCE approach requires one-sixth of the computational time of WAVECOUPLE. The SWAN run alone (uncou-

==pled) takes about 1/4 the computational time of the WAVEFORCE ROMS run.== Although this permits a greater malleability in the use of available machine power and time, it does not consider the coupling between the two models and should be ==viewed== as a compromise solution rather than optimal. The ability of resolving the general pattern of near-shore circulation, however, makes the WAVEFORCE an interesting approach for operational purposes.

## 5 5 Code and Data availability

The COAWST model source code and documentation are available trough the website:

$https://coawstmodel-trac.sourcerepo.com/coawstmodel_COAWST/.$

All data used in the present work as well as the operational model results are publicly available trough the PacIOOS website:

$http://oos.soest.hawaii.edu/pacioos/.$

10 *Acknowledgements.* Dr. Souza was supported by NOAA grant #NA07NOS4730207. Dr. Powell was supported by ONR grant #N00014-09-1-0939. We would like to thank Ning Li for providing the SWAN configuration and boundary conditions, and helping in the SWAN model description and evaluation.

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

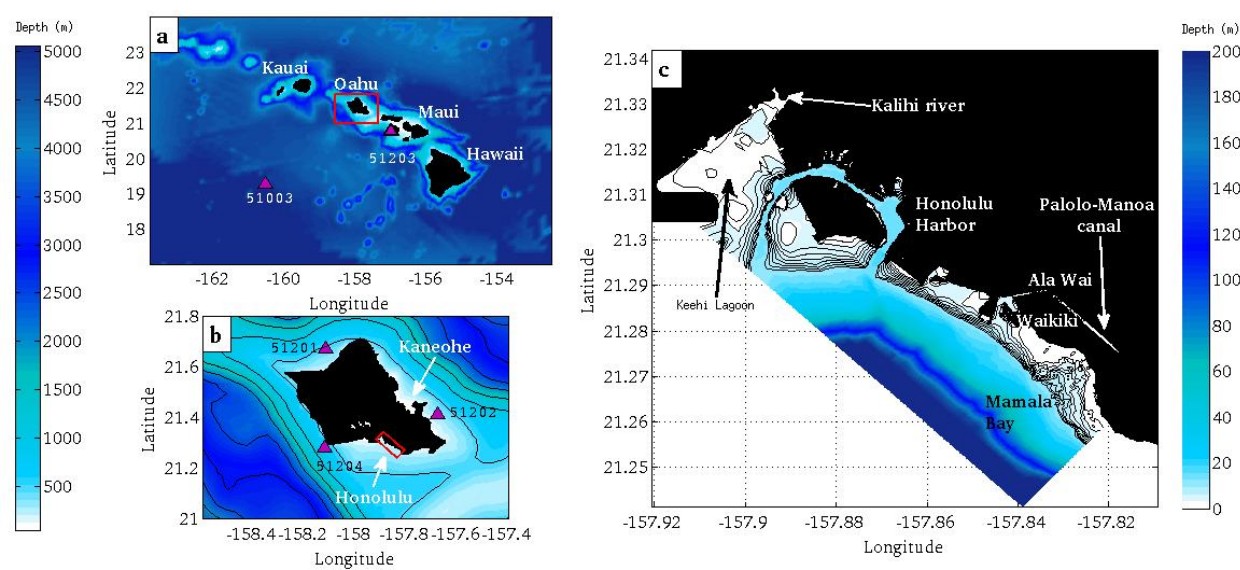

**Figure 1.** Bathymetry map of the (a) Hawaiian Islands, with the Island of O'ahu highlighted by the red rectangle and showed in detail in the map (b) (contour interval 500m). The numerical grid used in the present work /hl($\approx$50m resolution) is highlighted by the red rectangle in (b) and expanded in (c), with the black contours corresponding to the isobathymetric lines every 1m from the coast to the depth of 10m. It is possible to observe the intricate bathymetry near the coast of Honolulu associated to the coral reefs. The purple triangles indicate the positions of the NDCB buoys around Hawaii, that are used to validate the waves numerical model system.

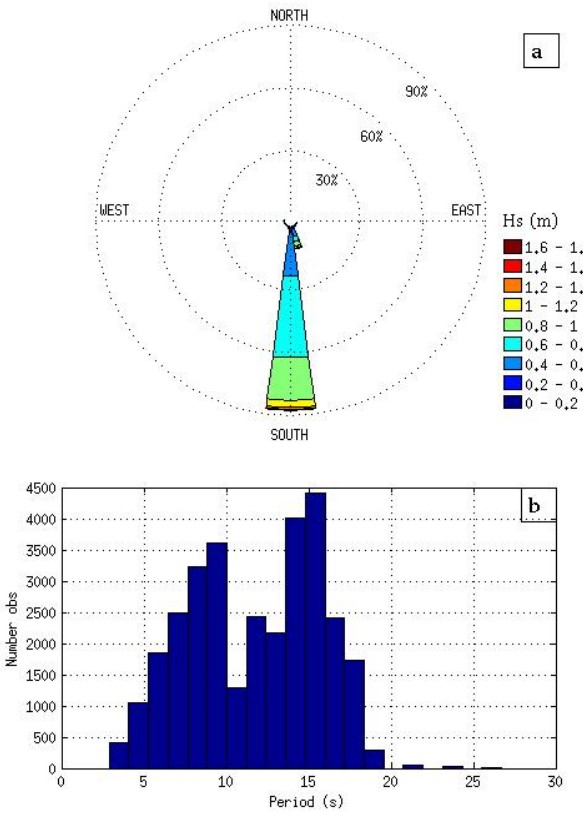

**Figure 2.** Wave conditions for the south shore of O'ahu as measured by the Kilo Nalu observatory. (a) shows the wave rose with frequency of occurrence (%) per wave significant height (Hs) and direction classes - the direction was binned in $10°$ classes, and (b) the histogram of wave peak period. One can see a large prevalence of waves from the south, with Hs usually under 1m, and two main peak periods around 10 and 15s.

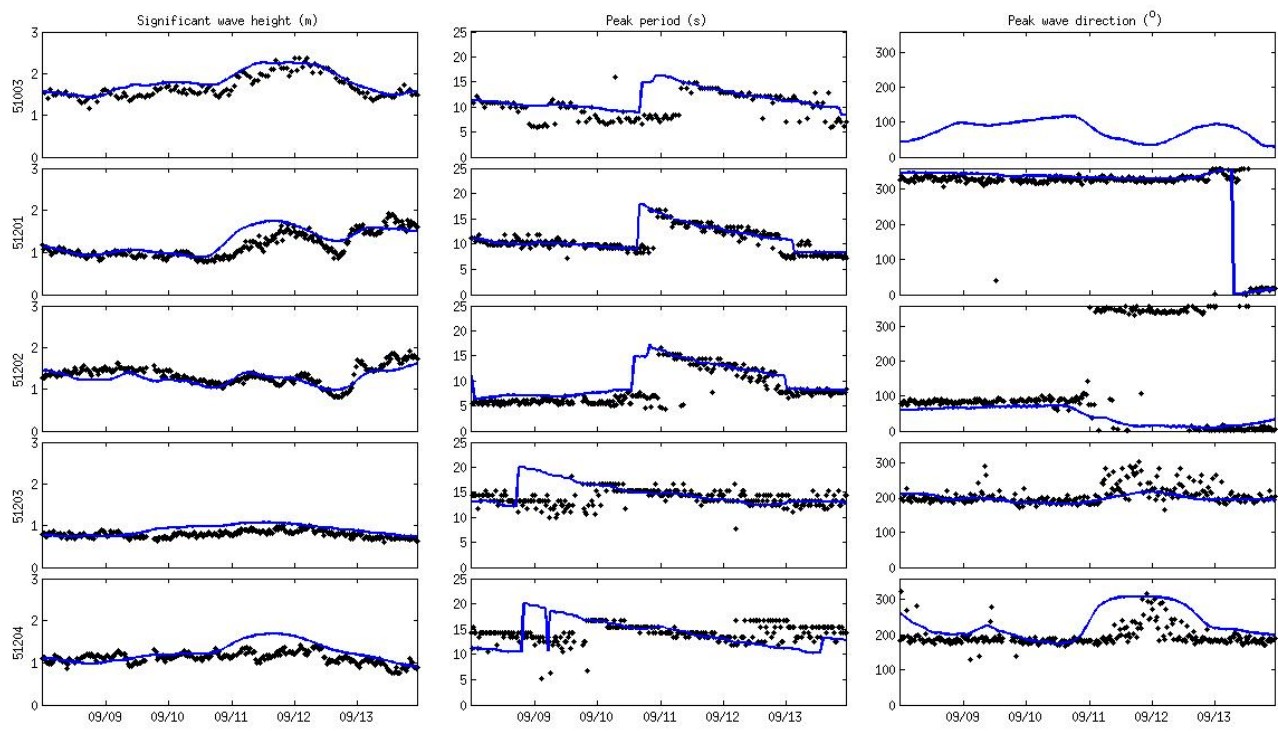

**Figure 3.** Comparison of the significant wave height, peak period and direction between the NDBC buoys dispalyed in Figure 1 and the wave model hindcast system results for the period of Experiment 1. The black dots correspond to the buys data and the blue line to the model results.

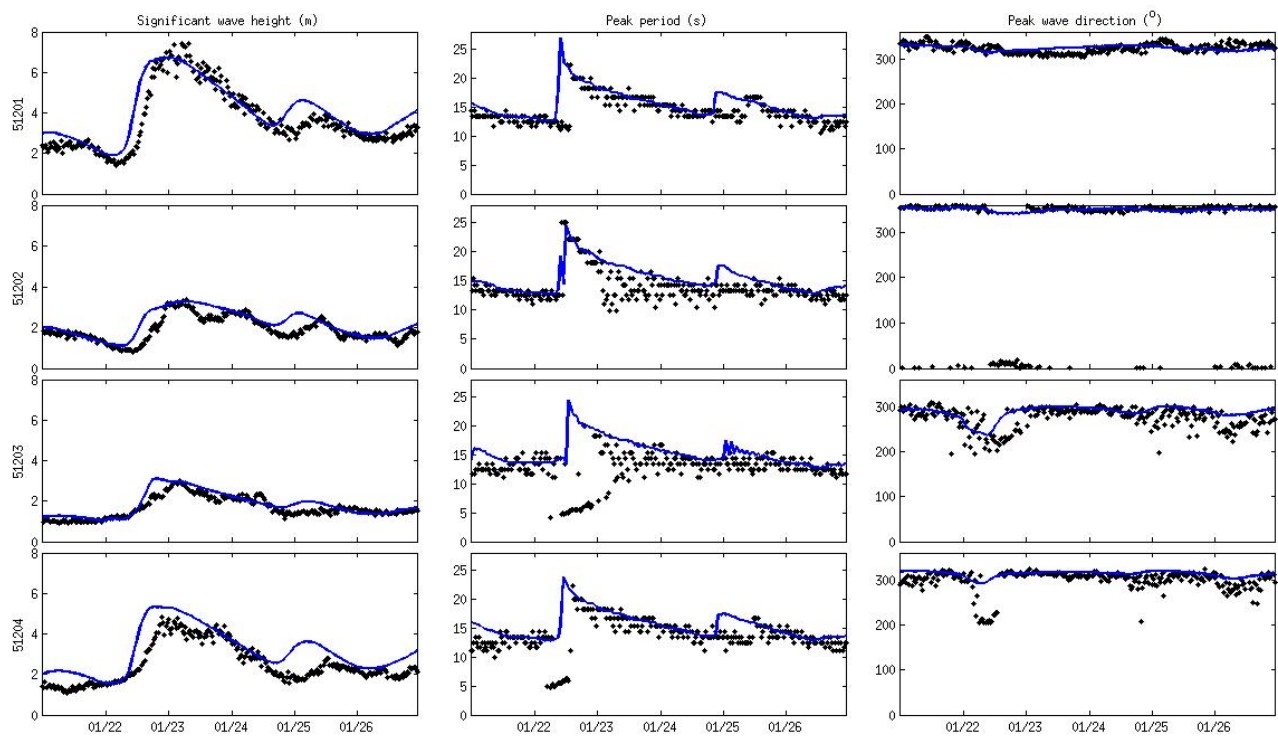

**Figure 4.** Comparison of the significant wave height, peak period and direction between the NDBC buoys dispalyed in Figure 1 and the wave model hindcast system results for the period of Experiment 2. The black dots correspond to the buys data and the blue line to the model results.

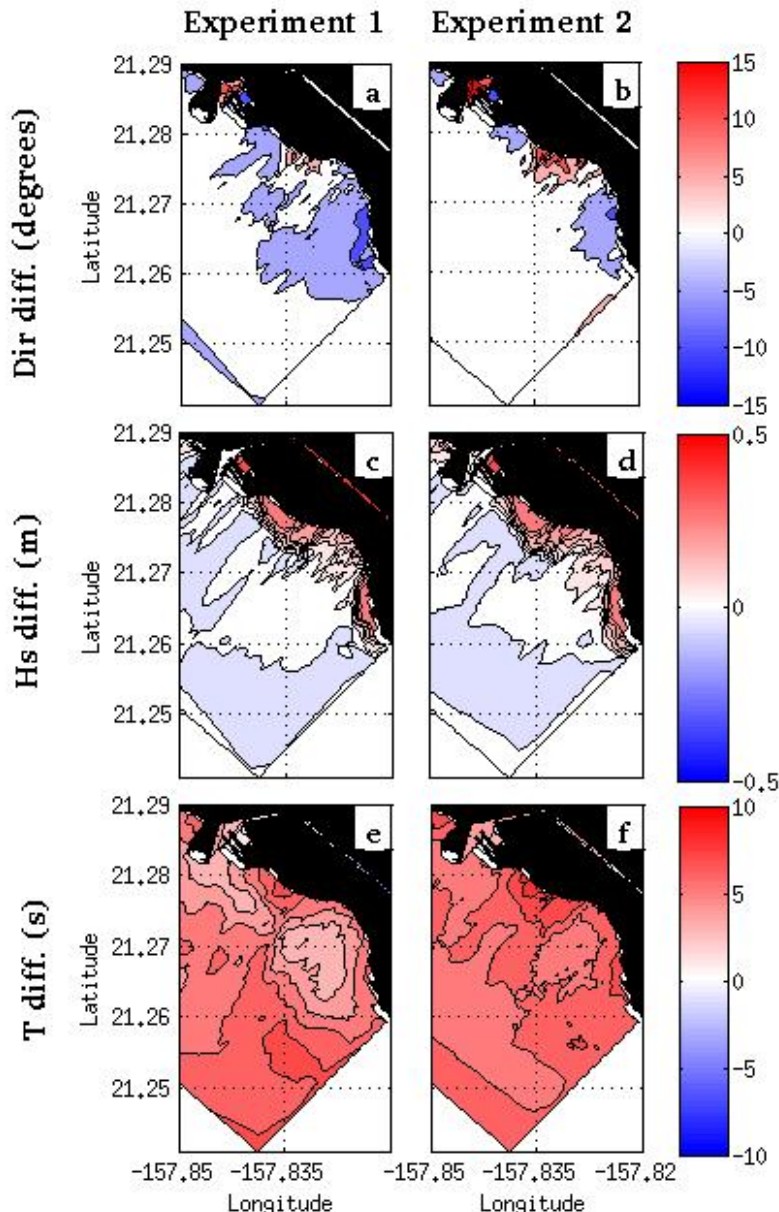

**Figure 5.** Maps of differences in wave direction (a, b - unit: ° from N - c.i. 5°), wave significant height Hs (c, d - unit: meters - c.i. 0.05m) and wave period T (e, f - unit: seconds - c.i. 1s), between the WAVECOUPLE and WAVEFORCE simulations (WAVECOUPLE - WAVEFORCE). Comparisons for the experiment 1 are on the left column and for experiment 2 on the right column. The coupled simulation presents significant higher Hs near the shore and larger T throughout the model domain for both experiments.

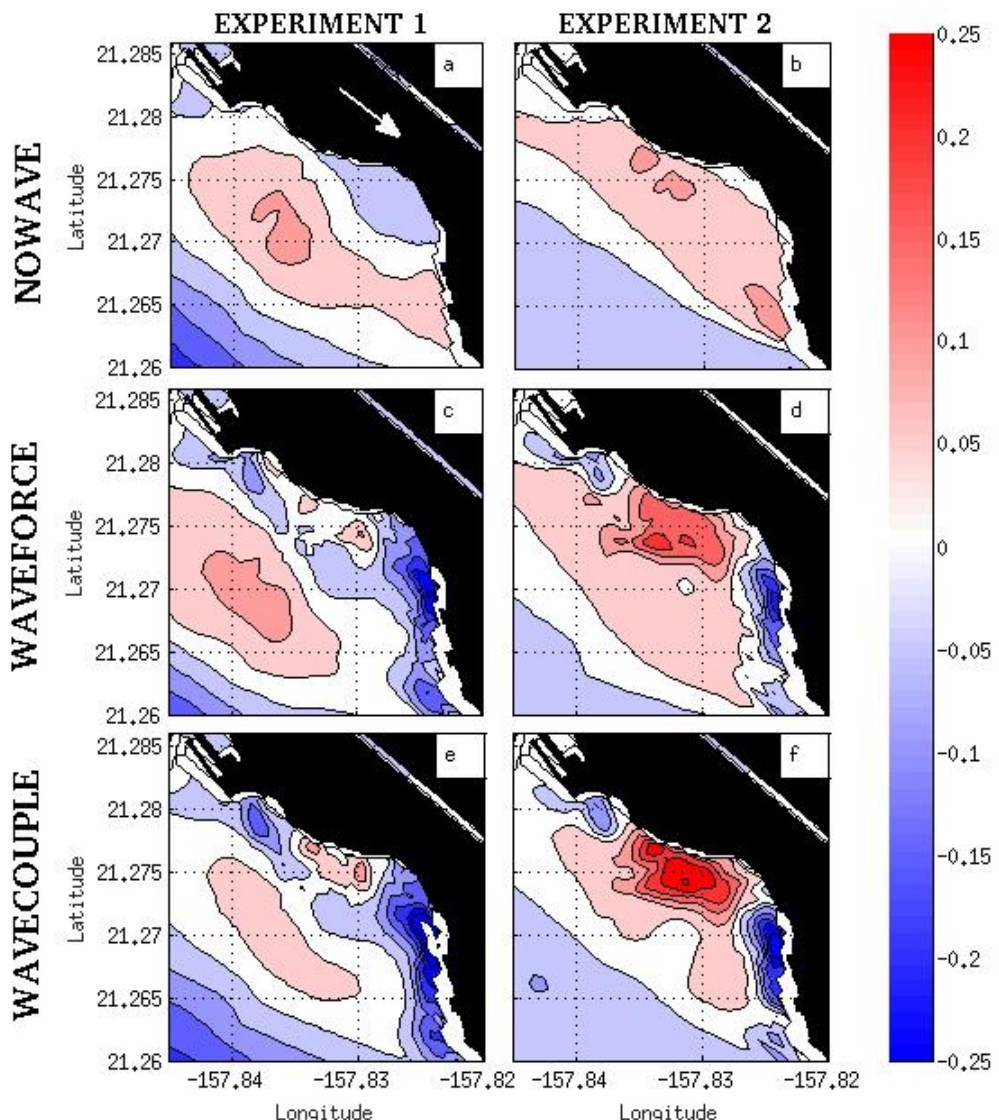

**Figure 6.** Time-averaged along-shore currents (m s$^{-1}$) in Waikiki region for the experiment 1 (a, c and e) and experiment 2 (b, d and f) from the 3 modeling strategies. The colors indicate the current intensity (m/s) with contours every 0.05 m/s. It is interesting to observe the appearance of drift currents along the coast in the simulations that consider the effect of the surface waves. Both the WAVEFORCE and WAVECOUPLE simulations resolve the modification of the near-shore circulation pattern by the waves, with WAVECOUPLE presenting stronger coastal drift currents. The white arrow in sub-figure (a) shows the direction of positive along-shore velocities.

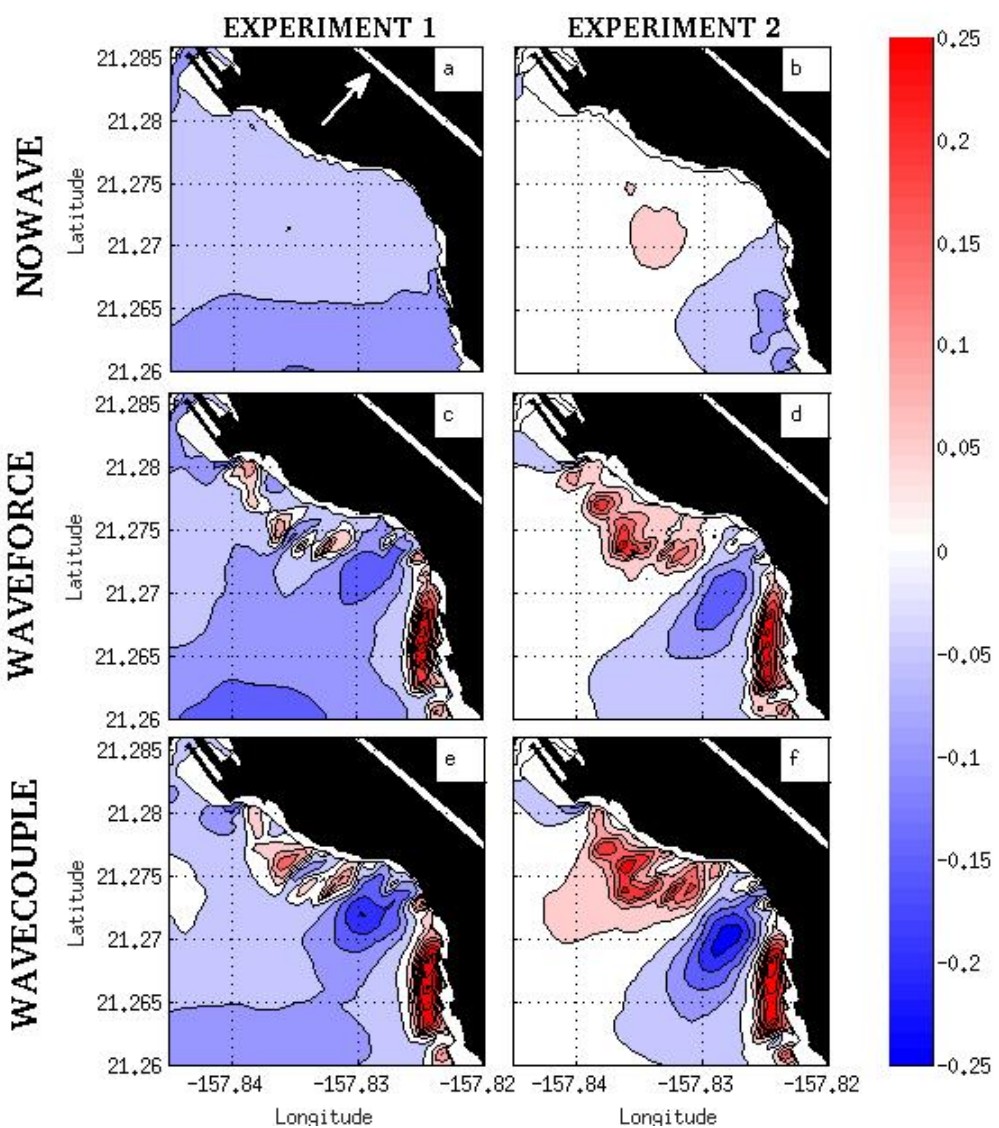

**Figure 7.** Time-averaged cross-shore currents (m s$^{-1}$) in Waikiki region for the experiment 1 (a, c and e) and experiment 2 (b, d and f) from the 3 modeling strategies. The colors indicate the current intensity (m/s) with contours every 0.05 m/s. It is interesting to observe the appearance of return flow cells perpendicular to the coast in the simulations that consider the effect of the surface waves. Both the WAVE-FORCE and WAVECOUPLE simulations resolve the modification of the near-shore circulation pattern by the waves, with WAVECOUPLE presenting stronger currents. The white arrow in sub-figure (a) shows the direction of positive cross-shore velocities.

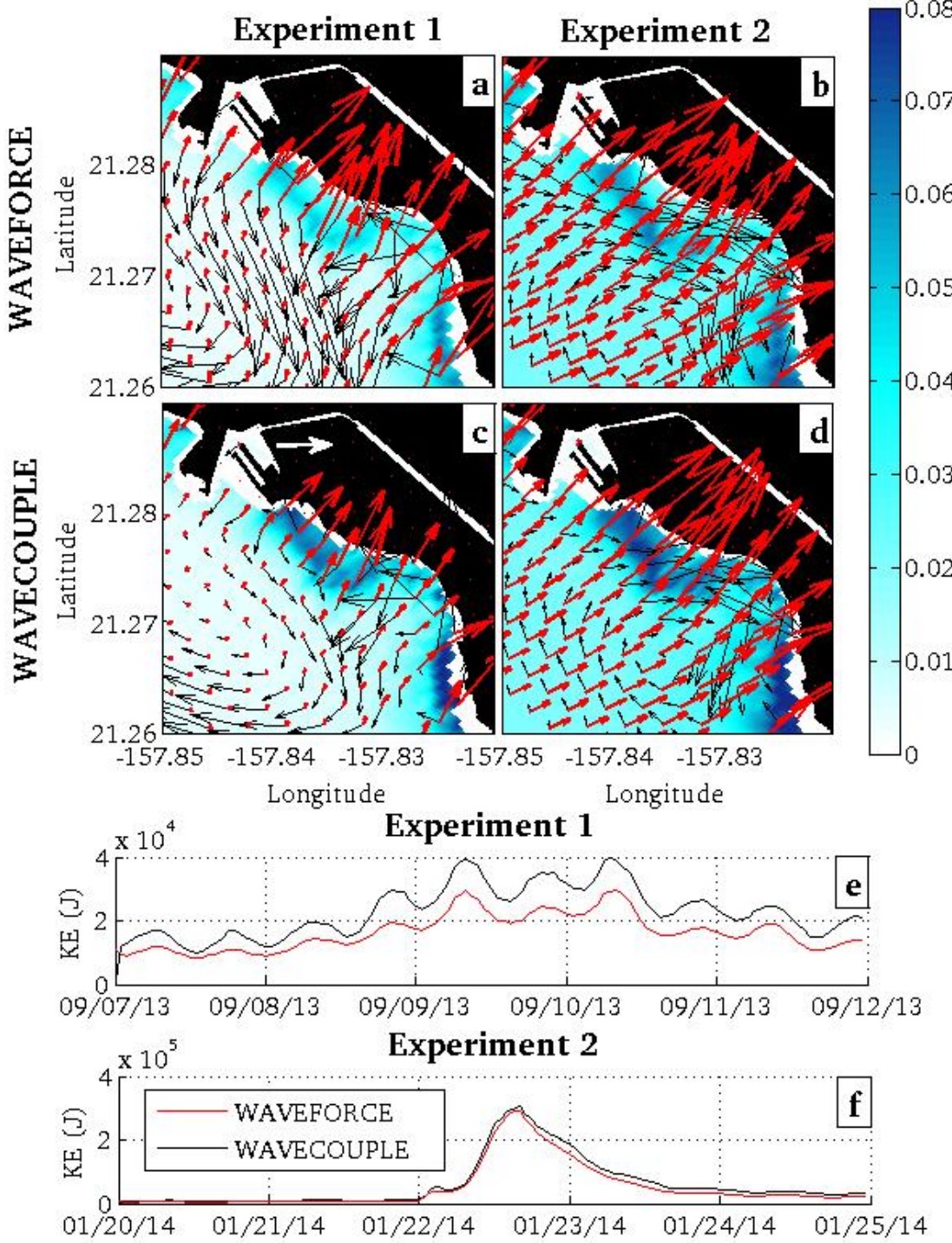

**Figure 8.** Maps of the time-averaged Stokes drift velocities (red arrows - m s$^{-1}$) and total surface velocities (black arrows - m s$^{-1}$) for the experiments 1 (a, c) and 2 (b, d), and the associated time series of Stokes drift velocities kinetic energy (e, f - J). The white arrow in panel c indicates a 10 cm s$^{-1}$ scale for the velocity vectors. Please note the difference in the vertical scale of panels e and f.

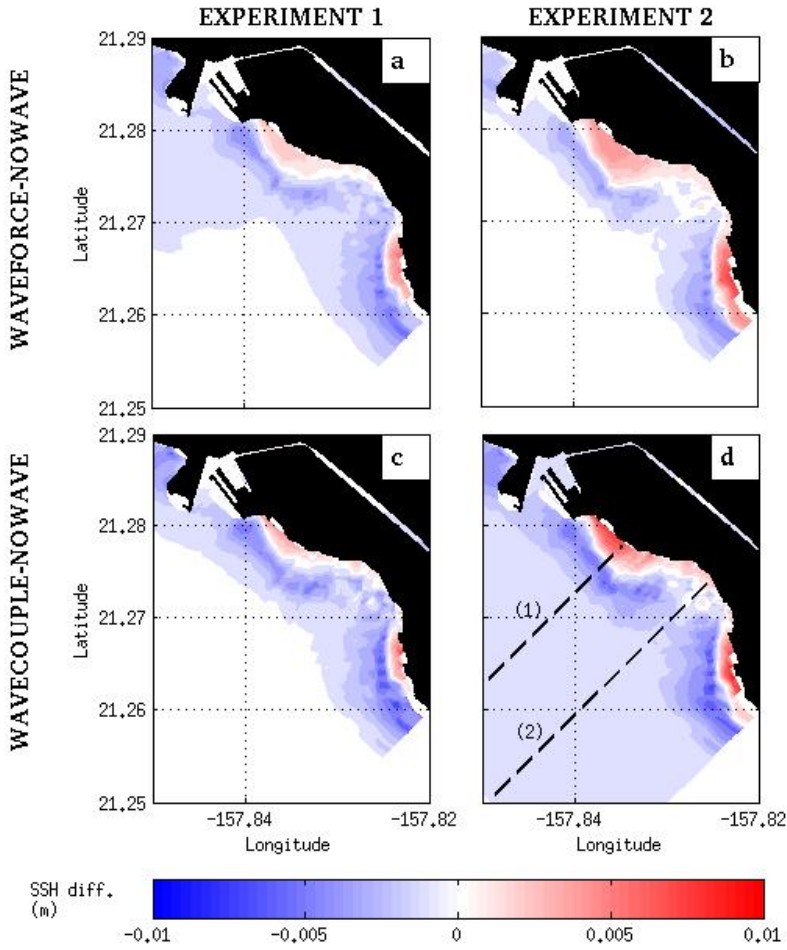

**Figure 9.** Time-averaged SSH difference (m) between the simulations that include the effect of surface gravity waves (WAVEFORCE and WAVECOUPLE) and the NOWAVE case for Experiment 1 (a, c) and Experiment 2 (b, d). The black dashed lines in (d) indicates the position of the 2 cross-shore sections used to analyze the wave setup and the cross-shore balance.

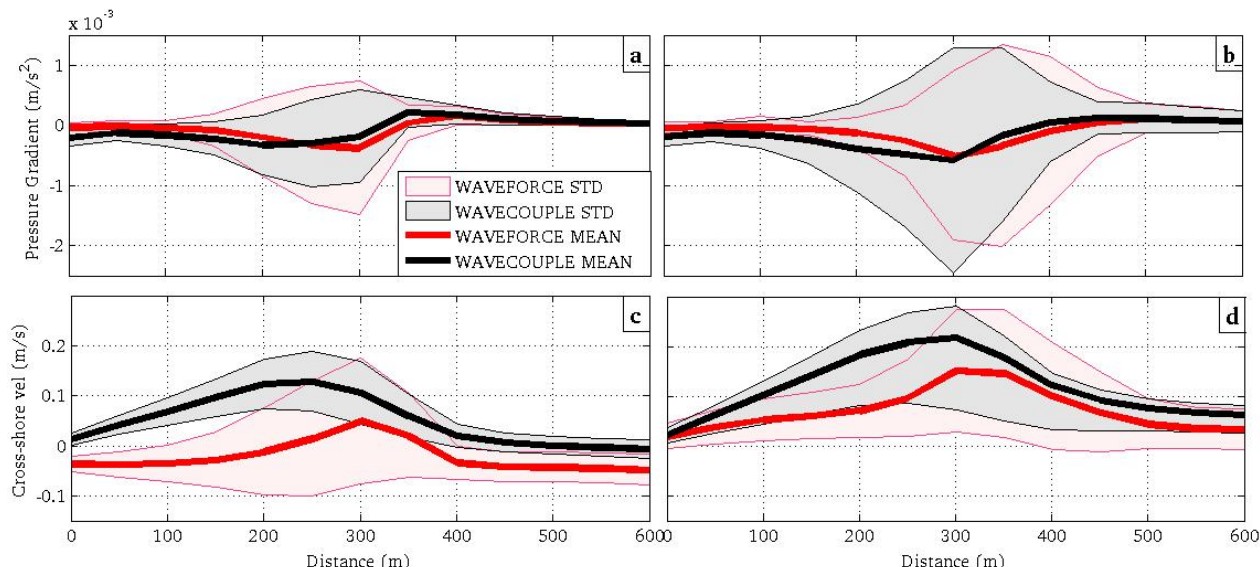

**Figure 10.** Sections of pressure gradient per density unit and surface cross-shore velocities for Experiment 1 (a, c) and Experiment 2 (b, d). The thick lines show the mean while the shaded areas are the standard deviations. Distances are measured from the coastline along the section defined in Figure 9d. Negative pressure gradient is directed offshore.

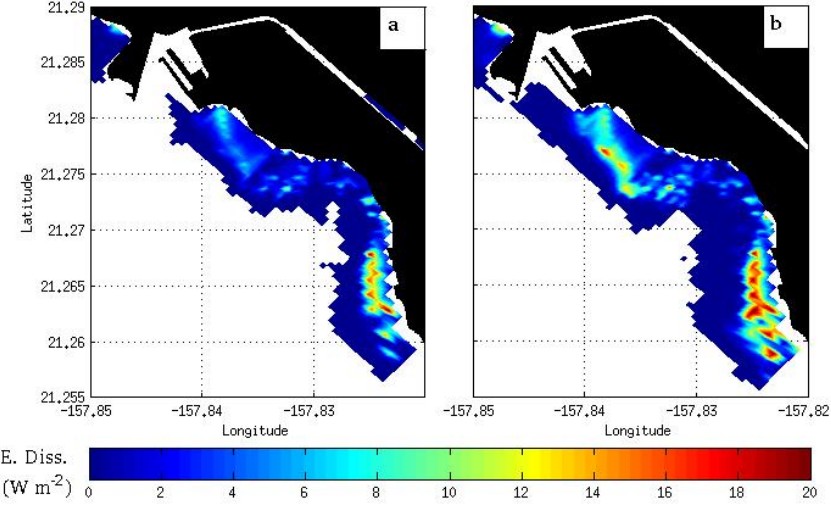

**Figure 11.** Maps of the time mean wave energy dissipation by depth induced wave breaking (W $m^{-2}$) for the WAVECOUPLE Experiment 1 (a) and Experiment 2 (b). The depth induced wave breaking was the most significant wave dissipation term. The energy dissipation was concentrated in the western and eastern portions of Waikiki due to the reef bathymetry (see Fig. 1). The larger dissipation in (b) is due to the large wave heights of Experiment 2

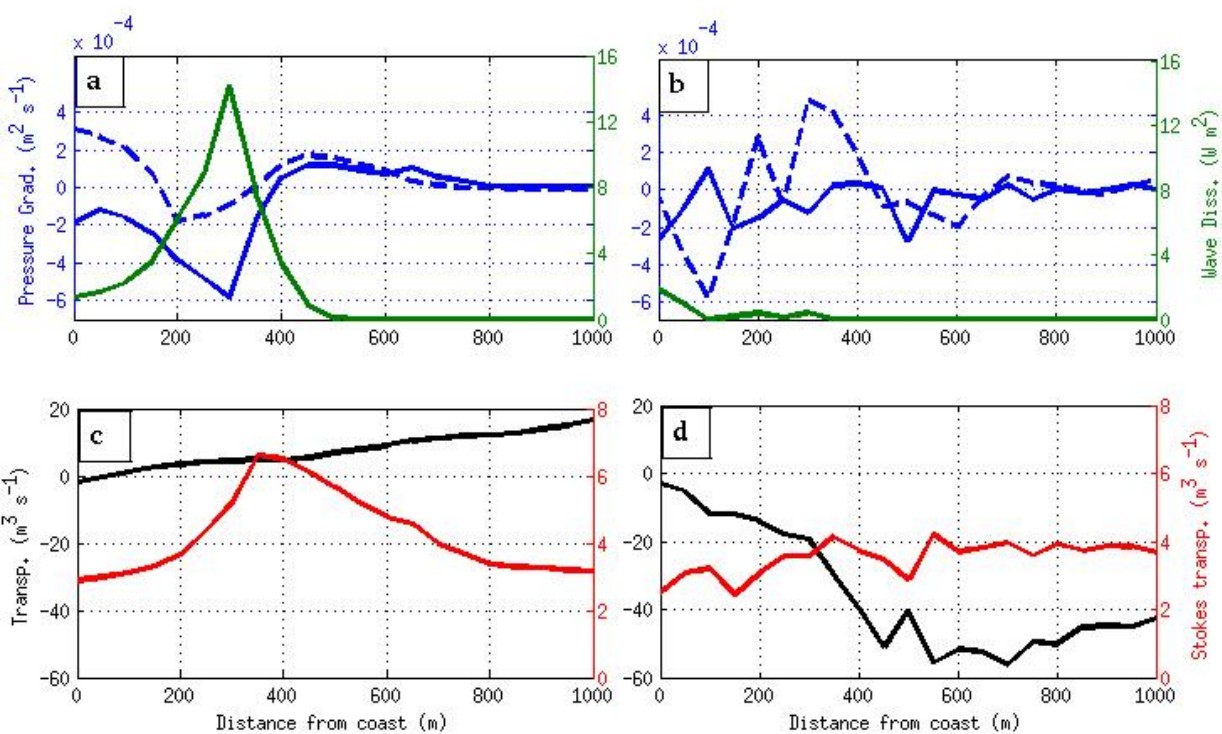

**Figure 12.** Cross-shore sections 1 (a, c) and 2 (b, d) of the wave contributions to the near shore momentum balance for the WAVECOUPLE Experiment 2. The section positions are shown in Fig. 9d. While section 1 is over the region of large wave breaking energy dissipation, the section 2 is in the region of the intense return flow observed in the cross-shore velocity map in Fig. 7. The pressure gradient (continuous line = cross-shore; dashed line = alongshore) due to wave setup is shown in blue, the wave dissipation due to depth induced wave breaking in green, the vertically integrated Stokes drift transport in red and the total wave induced cross-shore transport in black. The total wave induced cross-shore transport was obtained subtracting the NOWAVE quasi-Eulerian velocities from the WAVECOUPLE velocities.

**Table 1.** Root-mean-squared deviations for the wave significant height (Hs) and direction, between the model results and the NDBC buoys presented in Figure 1. All buoys are located in deep water, where the waves do not feel the bathymetry.

| Buoy ID | Experiment 1 | | Experiment 2 | |
|---|---|---|---|---|
| | Hs (m) | Dir. (°) | Hs (m) | Dir (°) |
| **51003** | 0.20 | – | – | – |
| **51201** | 0.18 | 18 | 0.61 | 26 |
| **51202** | 0.16 | 33 | 0.45 | 31 |
| **51203** | 0.11 | 21 | 0.29 | 28 |
| **51204** | 0.22 | 41 | 0.64 | 45 |

**Table 2.** Spatial mean correlation factors between the difference in wave direction and significant height (Hs) for the WAVECOUPLE and WAVEFORCE forced modeling strategies and the surface current intensity and direction. Only correlations significant to the 99% level were taken into consideration.

| | Wave Direction Diff. | | Wave Hs Diff. | |
|---|---|---|---|---|
| | Exp. 1 | Exp.2 | Exp. 1 | Exp. 2 |
| **Current Intensity** | 0.35 | 0.36 | 0.37 | 0.45 |
| **Current Direction** | 0.45 | 0.26 | 0.52 | 0.27 |

**Table 3.** Difference of the total and wave contributions to the surface velocities (%) between the WAVECOUPLE and WAVEFORCE simulations. The results show that while the contribution of the Stokes drift to the total velocities is higher by the same rate for both approaches, the same is not true for the wave setup contribution. A more complex interaction between waves and currents derived from the coupling gives rise to stronger near shore currents in the WAVECOUPLE simulations (positive values).

| | Experiment 1 | Experiment 2 |
|---|---|---|
| **Stokes drift** | 21 | 21 |
| **Wave setup/setdown** | 5 | 14 |
| **Total** | -4 | 4 |