# Peer review of "Different approaches to model the near shore circulation in the south shore of O'ahu, Hawaii"

_Ocean Science, 2016_

## Referee Comment (RC1) · Anonymous Referee #1 · 17 Oct 2016

General remarks:

This paper examines output from regional waves and circulation models of the south shore of O'ahu, Hawaii, for two typical scenarios. SWAN and ROMS are used to model the flow and wave fields, respectively, and their outputs are described in isolation, and when the models are coupled in a variety of ways, i.e. one- and two-way coupling. This way the differences due to wave-current interactions are described. The focus is on the difference in near shore processes in the region. The paper also discusses the operational suitability and feasibility of the different approaches.

I think the paper could be better structured to help the reader understand the content. One way to do this could be to have a model validation section, before the results,

detailing the validation of the models, i.e. the separate SWAN and ROMS models. It would also help if Sections 3.1 and 3.2 could be split up further. I was struck by the absence of validation of the ROMS model. I understand that the currents model cannot be validated to the extent that the waves model is in the paper, due to the scarcity of data. But, is there not even any water elevation records within the fine scale domain that could be presented, to show that the phase of the tide etc. is correct?

It would be useful to have some presentation of the baseline spatial variability of the wave parameters, such as Hs and Tp and wave direction, i.e. i.e. in the format of Fig. 4. This would allow the reader to gain some idea of the typical conditions, before being presented with the differences between the models (Fig. 4). Following on from this the differences in significant wave height (Hs) of around 0.5 m presented for Keehi Lagooon seem quite large for a "stagnant region". If this is the difference between two models, then what would Hs predictions be for each individual model?

I found the use of the phrase "experiment" in relation to modelled scenarios a bit strange. You could consider using "scenario" instead, but I understand this is probably a question of style and personal preference.

In Section 2.1 some more details of the tidal boundary forcing should be provided. For example, does the tidal forcing simply compose of waters elevations along the boundary, or are tidal velocities through the water column also used? Do the outer models (which this model is nested in) also provide tidal boundary forcing, i.e. are the models tidal, or has the tide been removed from the output of these models? Given that the forcing from the outer models is provided every 3 hours, I assume that they have no tide in them, but this should be clarified in the text.

Specific comments:

Page 4, line 17: It sounds like you are using a curvilinear grid. If this is the case please indicate this in the text.

[Figure]

Page 4, line 21: Regarding the percentage of the grid being deeper than 50 m, and it being deepest at the southern boundary. How deep is the model at the southern boundary?

Page 4, line 23: This text makes it sound like the ROMS model is forced with surface gravity waves. I understand the SWAN model, which is described later, to be the waves model for all the simulations. This needs to be explained here, as it is confusing to the reader.

Page 5, line 1: I don't understand how tidal forcing can be imposed as a "separate spectral forcing". This should be further explained, as I don't think this kind of forcing is that common. Do you mean harmonic forcing, i.e. tidal harmonic constituents are used to force the tide?

Page 6, line 30: please explain why this assumption about the wave field is necessary

Page 7, line 13: This sentence doesn't really make sense. You mention the difference between the wave parameters, but these differences haven't been presented yet, and you do not explain how these differences don't impact the model solution, or what this means. It is also not clear what the times in brackets are indicating.

Page 7, line 18 – page 8, line 2: This is a description of the validation of the waves model. You could consider having a separate subsection detailing the validation in order to help structure the paper a little more (see earlier comment about paper structure). It appears to me that these model results are from a SWAN only run, is that correct? This should be made clear. One way to do this would be to introduce another 'group of simulations', i.e. (4) WAVEONLY or similar.

Page 7, line 31 – page 8, line 2: Regarding the model representing the typical multi-modal wave conditions. I don't understand how this can be deduced from the results presented so far can, i.e. a comparison of wave parameters with measurements (Figs 2 and 3). I understand multi model seas to be hard to represent with parameters such

a s peak period, and that an analysis of the wave spectrum is necessary. Could this be done? Or could more of an explanation about how you came to your conclusion be provided? It would also be useful for the multi-model nature of experiment 1 to be mentioned earlier, in section 2.4.

Page 8, line 3: You should supply results / evidence to support this statement, i.e. introduce Fig 4 here and say how this shows this. Fig 4 shows how Hs is lager, but there's not presentation of wave periods.

Page 8, line 23: You mention that ∼20% higher Hs if considering the whole domain. Is this because of the contribution at Keeki Lagoon?

Page 8, line 29: You mention the return flow area in the middle of the beach. It is hard to discern the beach in the figure. I can see an area of no change in Hs, and this must be this section of the beach. It would be useful to have the extent of the beach indicated somehow; maybe in Fig 1 if that's easier.

Page 8, lines 31- 34: Can you provide some more details as to how the correlation analysis was performed.

Page 9, lines 18 – 21. Please explain how the along-shore and cross-shore components were calculated.

Page 13, lines 31 -34: The reduction in computation cost of WAVEFORCE, compared to WAVECOUPLE is discussed. Surely the waves model alone has some computational cost and it would be interesting to mention this, as this would also have to be run in an operational modelling scenario.

Page 13, line 27: can you consider specifying the percentage increase in Hs and Tp in brackets.

Minor Points and typos:

Fig1c should have a label and units for the colour scale (depth).
Page 6, line 28: "thickness" -> "depth"

Page 6, line 30: "what was found . . ." -> "which was found . . ."

Page 7, line 4: "These simulations . . ." -> "Each of these simulations . . ."

Page 8, line 7: The current, u, should be defined in the text

Page 8, line 18: What the difference is Hs is should be stated, i.e. is it Hs_(WAVEFORCE) – Hs_(WAVECOUPLE) or the other way around?

Page 8, line 25: "/reff4B"

Page 9, line 5: "prevalence of small waves (. . .), this emphasizes . . ." -> "prevalence of small waves (. . .) emphasizes . . ."

Page 10, line 13: "associated to the . . ." -> "associated with the . . ."

Page 10, line 15: "observed differences in the total currents between the WAVEFORCE AND WAVECOUPLE models"

Fig 8: The first line of the caption doesn't quite make sense.

Page 11, line 32: Fif. /reff11

Page 12, line 10: "as evident" -> "as is evident"

Page 12, line 14: "seams" -> "seems"

Page 12, line 33: "aim on providing" -> "aim to provide"

Page 13, line 25: "with general stronger" -> "generally with stronger"

Page 14, line 2: "view" -> "viewed"

---

## Short Comment (SC1) · 18 Oct 2016

First of all, thank you for the thoughtful review of our manuscript. All your comments are being taken into consideration for the revised manuscript. There is one point, however, I would like to discuss a little further. I have been dedicating a lot of thought to the validation of the circulation model. For the "scenarios" time period there are no observations that i know of. The only level obs are inside Honolulu Harbour and are not representative of the ocean tide amplitudes an phase (https://tidesandcurrents.noaa.gov/stationhome.html?id=1612340) - that makes it tricky to compare to a model that doesn't fully resolve the harbour channel. I would really appreciate if you (or anyone) have any information on datasets on the near-shore circulation, water levels or whatever other data that can be used to validate our runs. That said, we believe that comparing different model results can be useful to give us some insight on the circulation patterns and near-shore process in such an important region.

Kind regards

Joao Souza

---

## Referee Comment (RC2) · Anonymous Referee #2 · 25 Oct 2016

General Comments:

This paper explores the nearshore circulation differences that arise from forcing the ocean model with wave model output (one-way) compared to coupling the ocean and wave models together (two-way). For this comparison, the authors chose to model the south shore of O'ahu, building on a model used in several previous studies.

I was struck by the lack of statistics and model/observation comparisons used to evaluate the simulations. I recognize that comparing these two methods (one-way and two-way) of incorporating waves into the model with each other and using the ocean model only simulation as the base case will reveal robust features in the circulation and differences in the circulation due to wave-current interactions and model coupling. The

authors state there were no observations in the study area during the two experiments but reference a study using the lower resolution model system that was compared to satellite observations. I appreciate this reference but think expanding it to include more information (e.g. RMSD) would be valuable. Also, the term "validate" is used throughout the manuscript. I think "evaluate" is more accurate because no model perfectly reproduces the circulation, but recognize that this is my own personal preference.

Specific Comments:

Title: Oahu vs O'ahu – "O'ahu" is mostly used in the body of the paper, whereas "Oahu" used in the title and a few times in paper. Please be consistent throughout.

Section 2.1: Provide more details about the boundary conditions. You specifically mention the southern boundary of the high resolution nest, but what about the other boundaries? The manuscript says both "The southern boundary is forced by the barotropic tide, surface gravity waves, and the circulation from the coarser..." and "Eleven tidal constituents ... were introduced as a separate spectral forcing in the outer grids". Please clarify if tidal forcing was applied to the lower resolution grids and then propagated through the boundary conditions into the higher resolution grid, or if the high resolution nest included tidal forcing whereas lower resolution grids did not.

Page 4, line 19: I like your description of the vertical layers and domain depth. Please add the ROMS model minimum depth.

Page 4, line 23: "...coarser 200m parent-grid" does not seem to match "ROMS circulation models of approximately 250m, 1km, and 4km resolutions" (page 4, line20).

Page 4, line 23: "forced by... surface gravity waves" – Are the surface gravity waves in the ROMS model or in SWAN and then coupled using MCT?

Section 2.2: Please state the SWAN grid domain used in this application. Which resolution ROMS grid is used, or is a new grid of a different domain/resolution used for the SWAN simulation? If a different domain/resolution was used, please explain this

choice.

Page 13, line 28: "...significant improvements to the coupling..." – how do you know the coupling improves the simulation without comparisons of the ocean/wave model simulations to any observations? I assume that the two-way coupled solution is better, and I think other papers have reached that conclusion as well. You have nicely expressed how the WAVEFORCE and WAVECOUPLE solutions differ, but to show improvement I think observational comparisons may be necessary or describe how the coupled solution improves from the wave-forced simulation.

Minor Issues/typos:

Page 2, line 25: "Each of these issues can be significant..." I think you want another word than "issues" - perhaps features, phenomena or factors?

Page 2, line 28: "...circulation/waves model..." –> "... circulation/wave model..."

Page 3, line 25: "interaction in the south shore" –> "interaction off the south shore"

Page 6, line 30: "... what was found..." –> "... which was found..."

Section 2.2: Include statistical comparison of the SWAN model output compared to NDBC 5 buoys in the area (and/or with Figure 2). Does the SWAN grid cover the entire model domain or only the high resolution South Shore of O'ahu nest?

Section 2.3: Is the WAVECOUPLE case, is the two-way feedback only the highest resolution nest? Page 7, lines 20-25: "Figure 2 shows good agreement between the measured and modeled..." –include statistical comparison.

Page 9, line10: "Oahu for the experiments period" –> "Oahu during the experiment's period"

Page 11, line 28: "stokes drift" –> "Stokes drift"

Page 11, line 33: "Fig./reff11."

Page 13, line 31: "when aiming on resolving" –> "when aiming to resolve"

Page 14, line 2: "should be view as" –> "should be viewed as"

Figure Comments:

Figure 1: Label the colorbars, and include grid resolutions in caption.

Figure 2/3: Peak wave direction units: degrees (from true north, or east?). Include statistical comparison.

Figure 4: Since the domain for each subplot is the same, tick label latitudes only on the left column and tick label longitudes only on the bottom row. Include a bold row label (similar to Figure 5) for "Experiment 1" and "Experiment 2" on the left side for clarity. Or to be consistent with future plots that have Experiment 1 in column 1 and experiment 2 in column 2 and use bold label for the property being plotted (direction or Hs) and respective colorbar on right. For consistency with other figures, corner subplot labels (a,b,c,d) could be white boxes, but this is personal preference.

Figure 5: Since the domain is the same for each subplot, tick label latitudes only on left subplots and tick label longitudes only on the bottom row. Label experiment on the top of each column. Label the colorbar.

Figure 6: same suggestions as figure 5.

Figure 7: same suggestions as figure 5. Also, the plotted field for (a-d) is not labeled by the colorbar or stated in the figure caption. Also, include reference vectors for Stokes drift and total surface velocity. Subplots (e,f) include KE units and may want to note the order of magnitude difference (104 vs 105).

Figure 8: same suggestions as figure 5. For continuity, make the "waveforce-nowave" labels same format as previously. Also for continuity, place colorbar to the left of subplots and label.

Figure 9. Since x and y tick labels are the same, only label y-axis ticks on the left

column and x-axis ticks on the bottom row. X-label "distance from coastline (m)". Add Experiment 1/2 column labels. Consider flipping the x-axis so that distance starts at zero on the right and moves offshore, which is similar to the horizontal map. May consider making the "zero-line" bold or more distinct.

Figure 10 caption: "breaking (W m−2 )for the" –> "breaking (W m−2 ) for the" (add space)

Figure 10: Only label y-ticks on the left plot. For continuity, may consider plot labels (a,b) in northeast corner like other plots.

Figure 11: Consider flipping the x-axis so that distance starts at zero on the right and moves offshore, which is similar to the horizontal map. Move subplot labels (a,b,c,d) to the northeast corner for continuity with other figures. Remove x-axis tick labels of the top row. May consider removing y-ticks in the middle between plots, since both y-axes are the same.

---

## Author Comment (AC1) · 2 Dec 2016

General remarks:

This paper examines output from regional waves and circulation models of the south shore of O'ahu, Hawaii, for two typical scenarios. SWAN and ROMS are used to model the flow and wave fields, respectively, and their outputs are described in isolation, and when the models are coupled in a variety of ways, i.e. one- and two-way coupling. This way the differences due to wave-current interactions are described. The focus is

on the difference in near shore processes in the region. The paper also discusses the operational suitability and feasibility of the different approaches. I think the paper could be better structured to help the reader understand the content. One way to do this could be to have a model validation section, before the results, detailing the validation of the models, i.e. the separate SWAN and ROMS models. It would also help if Sections 3.1 and 3.2 could be split up further. I was struck by the absence of validation of the ROMS model. I understand that the currents model cannot be validated to the extent that the waves model is in the paper, due to the scarcity of data. But, is there not even any water elevation records within the fine scale domain that could be presented, to show that the phase of the tide etc. is correct? It would be useful to have some presentation of the baseline spatial variability of the wave parameters, such as Hs and Tp and wave direction, i.e. in the format of Fig. 4. This would allow the reader to gain some idea of the typical conditions, before being presented with the differences between the models (Fig. 4). Following on from this the differences in significant wave height (Hs) of around 0.5 m presented for Keehi Lagooon seem quite large for a "stagnant region". If this is the difference between two models, then what would Hs predictions be for each individual model? I found the use of the phrase "experiment" in relation to modelled scenarios a bit strange. You could consider using "scenario" instead, but I understand this is probably a question of style and personal preference. In Section 2.1 some more details of the tidal boundary forcing should be provided. For example, does the tidal forcing simply compose of waters elevations along the boundary, or are tidal velocities through the water column also used? Do the outer models (which this model is nested in) also provide tidal boundary forcing, i.e. are the models tidal, or has the tide been removed from the output of these models? Given that the forcing from the outer models is provided every 3 hours, I assume that they have no tide in them, but this should be clarified in the text.

Author: First we would like to thank the reviewer for the precise and complete review of our manuscript. We made a few changes to the manuscript structure, adding a subsection on model validation where we bring some information on previous efforts

to evaluate the model system results. Unfortunately, there are no observations to the best of our knowledge on the near shore circulation in O'ahu. The only water level measurement is obtained inside the harbor channel – that is not properly resolved by our model – and can not be used for validation. Since the outer grid assimilate High Frequency Radar surface velocities in the vicinity of the near shore domain, the currents provide as boundary conditions have very small phase shift (∼5min) in relation to the real measured currents. Due to the small domain covered by the inner grid, we believe there will be no significant change in the phase – although we cannot prove it without observations. A more detailed explanation on how the model include the tides is given. A new figure (Figure 2) was included presenting the baseline of wave conditions as measured by the Kilo Nalu observatory. The Keehi Lagoon is not the focus of the present analysis and was removed from the figures. That said, we believe our model didn't resolve the complete geography and bathymetry of the Lagoon and the interaction of the tidal currents in the channels with the wind waves.

Specific comments:

Page 4, line 17: It sounds like you are using a curvilinear grid. If this is the case please indicate this in the text.

Author: The grid is rectangular rotated to fit the shore orientation. Page 4 – Lines 19 to 22

Page 4, line 21: Regarding the percentage of the grid being deeper than 50 m, and it being deepest at the southern boundary. How deep is the model at the southern boundary?

Author: It varies. The deepest part in the southeast is 300m – and it decreases fast towards the cost as shown in Fig. 1. Page 4 – line 25

Page 4, line 23: This text makes it sound like the ROMS model is forced with surface gravity waves. I understand the SWAN model, which is described later, to be the waves

model for all the simulations. This needs to be explained here, as it is confusing to the reader.

Author: We modified this paragraph to make it clearer. In reality the ROMS does not receive waves boundary conditions - only SWAN. The WAVEFORCE ROMS simulation is forced with surface waves from an independent SWAN simulation. While in WAVECOUPLE the models are coupled. Page 4 - line 29

Page 5, line 1: I don't understand how tidal forcing can be imposed as a "separate spectral forcing". This should be further explained, as I don't think this kind of forcing is that common. Do you mean harmonic forcing, i.e. tidal harmonic constituents are used to force the tide?

Author: A complete explanation is given. The barotropic velocities and water level are given at the boundary, as well as tidal potential in each grid point. Page 5 – lines 8 to 13

Page 6, line 30: please explain why this assumption about the wave field is necessary

Author: This is not a necessary assumption – just the way the model system and in particular SWAN works (SWAN = Simulating Waves NEARSHORE). In reality, there are efforts to couple ROMS to other models without this limitation. The fact is that close to the shore this is a reasonable approximation. But if one goes to deep water the influence from several different swells coming from different directions should be taken into account.

Page 7, line 13: This sentence doesn't really make sense. You mention the difference between the wave parameters, but these differences haven't been presented yet, and you do not explain how these differences don't impact the model solution, or what this means. It is also not clear what the times in brackets are indicating.

Author: Thank you for point this out. There was a problem with the phrase formulation, that was corrected – we were referring to the communication time step between

models. Page 7 – line 28

Page 7, line 18 – page 8, line 2: This is a description of the validation of the waves model. You could consider having a separate subsection detailing the validation in order to help structure the paper a little more (see earlier comment about paper structure). It appears to me that these model results are from a SWAN only run, is that correct? This should be made clear. One way to do this would be to introduce another 'group of simulations', i.e. (4) WAVEONLY or similar.

Author: Yes, the evaluation of the wave is made on a SWAN run. This correspond to the WAVEFORCE simulations – where SWAN is run separate from ROMS. Page 8 – lines 3 to 23

Page 7, line 31 – page 8, line 2: Regarding the model representing the typical multi-modal wave conditions. I don't understand how this can be deduced from the results presented so far can, i.e. a comparison of wave parameters with measurements (Figs 2 and 3). I understand multi model seas to be hard to represent with parameters such as peak period, and that an analysis of the wave spectrum is necessary. Could this be done? Or could more of an explanation about how you came to your conclusion be provided? It would also be useful for the multi-model nature of experiment 1 to be mentioned earlier, in section 2.4.

Author: We were referring to the comparison with NDBC buoys we described earlier in the same paragraph. We modified the text to make it clearer. Page 8 – lines 5 to 7

Page 8, line 3: You should supply results / evidence to support this statement, i.e. introduce Fig 4 here and say how this shows this. Fig 4 shows how Hs is lager, but there's not presentation of wave periods.

Author: We added this result to the old figure 4 (now fig. 5). Figure 4

Page 8, line 23: You mention that 20% higher Hs if considering the whole domain. Is this because of the contribution at Keeki Lagoon?

Author: This was removed since we do not trust the results in Keehi Lagoon. Our model does not resolve the intricate channels that connect the lagoon to the ocean, with tidal plains. There are strong tidal currents in the model results of the lagoon, that should influence the Hs in the coupled runs.

Page 8, line 29: You mention the return flow area in the middle of the beach. It is hard to discern the beach in the figure. I can see an area of no change in Hs, and this must be this section of the beach. It would be useful to have the extent of the beach indicated somehow; maybe in Fig 1 if that's easier.

Author: We changed the phrase to refer to the small bay formed by the beach – that can be seen in the figures. We though this to be a better approach, since the map in fig. 1 has already a lot of text – what could make it confusing to indicate the beach. Page 9 – line 28

Page 8, lines 31- 34: Can you provide some more details as to how the correlation analysis was performed.

Author: Time series of the differences and current direction and intensity in each grid point in the domain covered in Figure 4 were extracted and the obtained correlations spatially averaged. Page 9 line 32 – page 10 line 1

Page 9, lines 18 – 21. Please explain how the along-shore and cross-shore components were calculated.

Author: We refer to mean along-shore and cross shore. That mean, along and perpendicular to the mean shore orientation. Although it does not correspond to the perfect along and cross shore components at each individual point at the coast, this projection reflects the mean circulation from a regional point of view. Page 10 lines 19 and 20

Page 13, lines 31 -34: The reduction in computation cost of WAVEFORCE, compared to WAVECOUPLE is discussed. Surely the waves model alone has some computational cost and it would be interesting to mention this, as this would also have to be run

in an operational modelling scenario.

Author: The processing time for the SWAN alone for the near shore domain is trifling compared to ROMS. The coupling is what takes time. Page 14 line 34 – page 15 line 1

Page 13, line 27: can you consider specifying the percentage increase in Hs and Tp in brackets.

Author: Added following the reviewer comment. Page 14 line 27

Minor Points and typos:

Author: All typos were corrected and figures modified according to the reviewer comments.

Fig1c should have a label and units for the colour scale (depth).

Page 6, line 28: "thickness" -> "depth"

Page 6, line 30: "what was found" -> "which was found

Page 7, line 4: "These simulations" -> "Each of these simulations

Page 8, line 7: The current, u, should be defined in the text

Page 8, line 18: What the difference is Hs is should be stated, i.e. is it Hs_(WAVEFORCE) – Hs_(WAVECOUPLE) or the other way around?

Page 8, line 25: "/reff4B"

Page 9, line 5: "prevalence of small waves ( ), this emphasizes " -> "prevalence of small waves () emphasizes"

Page 10, line 13: "associated to the" -> "associated with the

Page 10, line 15: "observed differences in the total currents between the WAVEFORCE AND WAVECOUPLE models"

Fig 8: The first line of the caption doesn't quite make sense.

Page 11, line 32: Fif. /reff11

Page 12, line 10: "as evident" -> "as is evident"

Page 12, line 14: "seams" -> "seems"

Page 12, line 33: "aim on providing" -> "aim to provide"

Page 13, line 25: "with general stronger" -> "generally with stronger"

Page 14, line 2: "view" -> "viewed"

---

## Author Comment (AC2) · 2 Dec 2016

General Comments:

This paper explores the nearshore circulation differences that arise from forcing the ocean model with wave model output (one-way) compared to coupling the ocean and wave models together (two-way). For this comparison, the authors chose to model the south shore of O'ahu, building on a model used in several previous studies. I was struck by the lack of statistics and model/observation comparisons used to evaluate the simulations. I recognize that comparing these two methods (one-way and two-

way) of incorporating waves into the model with each other and using the ocean model only simulation as the base case will reveal robust features in the circulation and differences in the circulation due to wave-current interactions and model coupling. The authors state there were no observations in the study area during the two experiments but reference a study using the lower resolution model system that was compared to satellite observations. I appreciate this reference but think expanding it to include more information (e.g. RMSD) would be valuable. Also, the term "validate" is used throughout the manuscript. I think "evaluate" is more accurate because no model perfectly reproduces the circulation, but recognize that this is my own personal preference.

Author: First of all we would like to thank the reviewer for the detailed comments. We made an effort to take them all into consideration and believe it greatly helped to improve the manuscript quality. We added information on the validation (or evaluation) of the outer domains. The inner domains, however, could not be evaluated due to the lack of observations. We agree with the reviewer, and the term validation was substituted by evaluation throughout the manuscript.

Specific Comments:

Title: Oahu vs O'ahu – "O'ahu" is mostly used in the body of the paper, whereas "Oahu" used in the title and a few times in paper. Please be consistent throughout.

Author: We now use only O'ahu, following the hawaiian spelling.

Section 2.1: Provide more details about the boundary conditions. You specifically mention the southern boundary of the high resolution nest, but what about the other boundaries? The manuscript says both "The southern boundary is forced by the barotropic tide, surface gravity waves, and the circulation from the coarser " and "Eleven tidal constituents were introduced as a separate spectral forcing in the outer grids". Please clarify if tidal forcing was applied to the lower resolution grids and then propagated through the boundary conditions into the higher resolution grid, or if the high resolution nest included tidal forcing whereas lower resolution grids did not.

Author: This information is added to the section 2.1. In fact, surface gravity waves are included in all open boundaries. The tides are forced as tide components in the barotropic velocites and water level at the open boundaries and as tidal potential at every grid point. 11 constituents are used for such. The TPXO model was used to obtain the tide harmonic constituents for the outer grids, and harmonic analysis of the outer grid results was used to generate the constituents for the near shore domain. Section 2.1 – Page 4 line 8 to page 6 line 3

Page 4, line 19: I like your description of the vertical layers and domain depth. Please add the ROMS model minimum depth.

Author: The minimum depth is 0.5m to account for the shallow reef areas. We didn't use any wet/dry scheme – maybe something to consider for future work. Page 4 lines 25 and 26

Page 4, line 23: "coarser 200m parent-grid" does not seem to match "ROMS circulation models of approximately 250m, 1km, and 4km resolutions" (page 4, line20).

Author: Corrected – 250m. Page 4 line 30

Page 4, line 23: "forced by surface gravity waves" – Are the surface gravity waves in the ROMS model or in SWAN and then coupled using MCT?

Author: We modified this sentence since ROMS is not forced with surface waves – SWAN is. In the WAVEFORCE case, the waves from an independent SWAN run are imposed as a separate forcing to ROMS. In the WAVECOUPLE case, SWAN and ROMS run together (coupled). Page 4 lines 29 and 30.

Section 2.2: Please state the SWAN grid domain used in this application. Which resolution ROMS grid is used, or is a new grid of a different domain/resolution used for the SWAN simulation? If a different domain/resolution was used, please explain this choice.

Author: The SWAN domain is exactly the same used for ROMS. Page 6 line 20

Page 13, line 28: "significant improvements to the coupling" – how do you know the coupling improves the simulation without comparisons of the ocean/wave model simulations to any observations? I assume that the two-way coupled solution is better, and I think other papers have reached that conclusion as well. You have nicely expressed how the WAVEFORCE and WAVECOUPLE solutions differ, but to show improvement I think observational comparisons may be necessary or describe how the coupled solution improves from the wave-forced simulation.

Author: We agree with the reviewer, and this comment came more from an impression of the general results – being speculation. That said, this comment was modified so it still express the care one should take with computational cost without the affirmation that one simulation is better. Page 14 line 28

Minor Issues/typos:

Author: All typos were corrected following the reviewer comments.

Page 2, line 25: "Each of these issues can be significant". I think you want another word than "issues" - perhaps features, phenomena or factors?

Page 2, line 28: "circulation/waves model" –> "circulation/wave model"

Page 3, line 25: "interaction in the south shore" –> "interaction off the south shore"

Page 6, line 30: "what was found" –> "which was found"

Section 2.2: Include statistical comparison of the SWAN model output compared to NDBC 5 buoys in the area (and/or with Figure 2). Does the SWAN grid cover the entire model domain or only the high resolution South Shore of O'ahu nest?

Author: The inner grid is exactly the same used for ROMS. The outer grid results evaluation was improved, and rmsd in relation to the NDBC buoys were included in a new table (Table 1). Page 8 lines 20 to 23.

Section 2.3: Is the WAVECOUPLE case, is the two-way feedback only the highest

resolution nest?

Author: Yes, all coupling (WAVEFORCE and WAVECOUPLE) are run only for the near shore domain.

Page 7, lines 20-25: "Figure 2 shows good agreement between the measured and modeled" –include statistical comparison.

Author: Rmsd information was added in Table 1.

Author: All the following typos were corrected and figures modified according to the reviewer comments. In some cases we opted for not adding the labels to the colorbars, when there was not enough space in the figure to add it without making the figure confuse. In these cases the unitis were clearly expressed in the caption.

Page 9, line10: "Oahu for the experiments period" –> "Oahu during the experiment's period"

Page 11, line 28: "stokes drift" –> "Stokes drift"

Page 11, line 33: "Fig./reff11."

Page 13, line 31: "when aiming on resolving" –> "when aiming to resolve"

Page 14, line 2: "should be view as" –> "should be viewed as"

Figure Comments:

Figure 1: Label the colorbars, and include grid resolutions in caption.

Figure 2/3: Peak wave direction units: degrees (from true north, or east?). Include statistical comparison.

Figure 4: Since the domain for each subplot is the same, tick label latitudes only on the left column and tick label longitudes only on the bottom row. Include a bold row label (similar to Figure 5) for "Experiment 1" and "Experiment 2" on the left side for clarity. Or to be consistent with future plots that have Experiment 1 in column 1 and experiment

2 in column 2 and use bold label for the property being plotted (direction or Hs) and respective colorbar on right. For consistency with other figures, corner subplot labels (a,b,c,d) could be white boxes, but this is personal preference.

Figure 5: Since the domain is the same for each subplot, tick label latitudes only on left subplots and tick label longitudes only on the bottom row. Label experiment on the top of each column. Label the colorbar.

Figure 6: same suggestions as figure 5.

Figure 7: same suggestions as figure 5. Also, the plotted field for (a-d) is not labeled by the colorbar or stated in the figure caption. Also, include reference vectors for Stokes drift and total surface velocity. Subplots (e,f) include KE units and may want to note the order of magnitude difference (104 vs 105).

Figure 8: same suggestions as figure 5. For continuity, make the "waveforce-nowave" labels same format as previously. Also for continuity, place colorbar to the left of subplots and label.

Figure 9. Since x and y tick labels are the same, only label y-axis ticks on the left